# CoMRes: Semi-Supervised Time Series Forecasting Utilizing Consensus Promotion of Multi-Resolution

**Yunju Cho, Jay-Yoon Lee**
Graduate School of Data Science
Seoul National University
1 Gwanak-ro, Gwanak-gu, Seoul 08826, Korea
{jyj0729, lee.jayyoon}@snu.ac.kr

## Abstract

Long-term time series forecasting poses significant challenges due to the complex dynamics and temporal variations, particularly when dealing with unseen patterns and data scarcity. Traditional supervised learning approaches, which rely on cleaned and labeled data, struggle to capture these unseen characteristics, limiting their effectiveness in real-world applications. In this study, we propose a semi-supervised approach that leverages multi-view setting on augmented data without requiring explicit future values as labels to address these limitations. By introducing a consensus promotion framework, our method enhances agreement among multiple single-view models on unseen augmented data. This approach not only improves forecasting accuracy but also mitigates error accumulation in long-horizon predictions. Furthermore, we explore the impact of autoregressive and non-autoregressive decoding schemes on error propagation, demonstrating the robustness of our model in extending prediction horizons. Experimental results show that our proposed method not only surpasses traditional supervised models in accuracy but also exhibits greater robustness when extending the prediction horizon. Code is available at this repository: https://github.com/yjucho1/CoMRes

## 1 Introduction

Time series forecasting has seen significant advancements in recent years, with research making substantial progress in capturing the complex dynamics inherent in time-series data (Zhou et al., 2022; Nie et al., 2023; Chen et al., 2024; Liu et al., 2024a; Challu et al., 2023; Das et al., 2023; Wang et al., 2024; Zeng et al., 2023). Due to the complex and non-stationary nature of real-world systems, observed time series often exhibit intricate temporal patterns, such as multi-scale dependencies and diverse fluctuations. To capture the multi-scale characteristics effectively, several advanced models structurally incorporate multi-scale features(Chen et al., 2024; Shabani et al., 2023; Wang et al., 2024; Challu et al., 2023). This approach generally increases model capacity, allowing for the learning of complex patterns across different scales through an increase in parameters. However, this increased capacity also increases the risk of overfitting, especially when training on small or low-diversity datasets. Addressing these challenges is essential for developing stable and reliable multi-resolution forecasting models.

In time series analysis, several data augmentation methods have been proposed to enhance model's generalization performance (Wen et al., 2021b; Iglesias et al., 2023; Debnath et al., 2021; Semenoglou et al., 2023). Despite their widespread use, labeling augmented data remains a significant challenge, particularly in forecasting tasks. This challenge arises because the augmented data may alter the underlying dynamics of the time series and often introduce noise. Consequently, assigning accurate labels for future predictions becomes difficult, as the augmented past values represent artificial, potentially noisy data rather than actual observations. Even slight shifts in trends or periodic patterns can lead to increased prediction errors, thereby limiting the direct applicability of augmented data in time series forecasting. However, the use of diverse and previously unseen data is essential to address the complex dynamics and time-varying heterogeneity of long-term time series data. Therefore,

reducing the risk of model overfitting and enhancing generalization ability without manually assigning future values to augmented data is crucial for effectively utilizing data augmentation in time series forecasting.

This study investigates the use of augmented data in long-term time series forecasting through a multi-view learning approach. Multi-view data has demonstrated considerable success in various domains, such as computer vision and natural language processing (Chen et al., 2021; Lim et al., 2020), by leveraging different manifestations of the same data to improve performance. In time series analysis, multi-view data can be represented by features such as varied temporal resolutions or distances, allowing for the extraction of temporal characteristics and dependencies across multiple time intervals, ultimately enhancing the accuracy and robustness of forecasts (Challu et al., 2023; Liu et al., 2022; Shabani et al., 2023; Chen et al., 2024). Many multi-scale modeling techniques integrate features from different time scales but are limited to labeled data. By employing semi-supervised learning, which leverages unlabeled data, we can fully utilize multi-scale features across diverse patterns.

Thus, we propose a semi-supervised approach that improves individual single-view models by enhancing consensus among them using unseen data. Inspired by traditional co-training methods (Blum & Mitchell, 1998; Lim et al., 2020), this approach leverages augmented unseen data to ensure that models not only capture multi-resolution patterns but also provide consistent predictions when faced with new, unseen data. The core of this **CoMRes** (**Co**nsensus promotion of **M**ulti-**Res**olution) lies in maximizing agreement among multi-view models on unseen data, thereby improving the overall performance and reliability of long-term time series forecasting.

In addition, we address the issue of error accumulation in long-horizon forecasting. In transformer-based research for long-term time series forecasting, a common approach is the use of a non-autoregressive decoding scheme, which projects future horizons all at once. Zhang et al. (2023) were the first to conduct a comparative evaluation of several models in terms of autoregressive and non-autoregressive decoding schemes. Motivated by their work, we evaluate our model with respect to both decoding styles. Specifically, we investigate how non-autoregressive models can be employed to minimize error accumulation when predicting values further into the future than the forecast range on which they were trained.

In summary, our contributions are as follows:

- We propose **CoMRes** (**Co**nsensus promotion of **M**ulti-**Res**olution), a novel semi-supervised time series forecasting that utilizes consensus promotion of multi-resolution information and augmented data without requiring explicit future values as labels to enhance long-term time series forecasting. To the best of our knowledge, this is the first work to investigate the influence of a semi-supervised framework in the context of long-term time series forecasting.

- We propose evaluating the forecasting model using both autoregressive and non-autoregressive decoding schemes. Our study focuses on how a forecasting model can extend its prediction horizon beyond the trained range while minimizing error accumulation, particularly when non-autoregressive models are used to generate autoregressive forecasts.

- To further understand CoMRes, we conduct ablation study the impact of incorporating data augmentation and consensus promotion strategies in semi-supervised and supervised learning.

- We evaluate the impact of our CoMRes under various training data size and analyze the effect of different data augmentation techniques to enhance model performance.

## 2 RELATED WORK

**Long-term Time Series Forecasting** Recent research has made significant progress in improving model architectures for time-series forecasting. Transformer based model such as FEDformer (Zhou et al., 2022) and Autoformer (Wu et al., 2021), apply attention mechanisms to multivariate time series data. These models address the quadratic complexity of traditional attention mechanisms by introducing novel mechanisms to reduce computational complexity. PatchTST (Nie et al., 2023) utilized patch-based representations to enhance local pattern recognition, while iTransformer (Liu et al., 2024a) and Crossformer (Zhang & Yan, 2023) pushed forward by capturing multivariate

correlations through sophisticated attention mechanisms. Meanwhile, several recent works (Das et al. (2023); Wang et al. (2023); Zeng et al. (2023); Wang et al. (2024)) demonstrate the effectiveness of non-transformer-based methods, offering competitive alternatives.

**Multi-scale Modeling for Time Series** Real-world time series often exhibit diverse variations and fluctuations across different temporal scales. To address this, recent research has increasingly focused on leveraging multi-scale information to better capture these complex temporal patterns. NHITS (Challu et al., 2023) employs multi-rate data sampling and hierarchical interpolation to model features of different resolutions. Pyraformer (Liu et al., 2022) and Scaleformer (Shabani et al., 2023) advanced the field with pyramidal attention mechanisms and iterative scale-refinement mechanisms, respectively, to capture multi-scale dependencies. TimeMixer (Wang et al., 2024) introduce a new multi-scale mixing architecture with decomposable mixing strategy. Pathformer (Chen et al., 2024) takes a more comprehensive approach by considering time series features from both different resolutions and temporal distances. Most previous methods integrate complementary forecasting capabilities through an ensemble of multiple predictions that rely on labeled data. By employing semi-supervised learning, which leverages unlabeled data, our approach effectively tackles the challenge of reconciling diverse representations, particularly in the context of unseen data.

**Time Series Data Augmentation** Comprehensive reviews of the most popular data augmentation approaches in time series analysis mostly focus on classification and anomaly detection tasks (Wen et al., 2021b; Iglesias et al., 2023). Data augmentation is easily applicable for time series classification and anomaly detection as small perturbations typically do not change the data label. Nonetheless, it is not straightforward to apply data augmentation techniques in time series forecasting as even small perturbations can lead to significant changes in the observations. Several studies (Debnath et al., 2021; Semenoglou et al., 2023; Zhou et al., 2023; Nochumsohn & Azencot, 2025) have examined the impact of data augmentation on forecasting, consistently finding that its effectiveness depends on data characteristics. Similarly, Wen et al. (2021a) survey time series augmentation methods, noting their benefits for classification and anomaly detection but reporting negative effects for certain data/model pairs in forecasting. Our findings (Table 1) align with these observations, showing that naive augmentation often degrades performance rather than improving it. Nochumsohn & Azencot (2025) propose an automated augmentation approach for long-term forecasting, iteratively searching across diverse time series transformations to optimize augmentation policies using Bayesian optimization. In contrast, our study applies simple augmentation solely for consensus promotion, achieving improvements in most experiments without requiring costly search procedures or complex augmentation techniques.

## 3 METHODOLOGY

In this paper, we study the problem of long-term time-series forecasting: Given historical data of a multivariate time series with a look-back window $L : (x_{t-L}, \ldots, x_t)$, where $x_t$ denotes the observation at timestamp $t$, our objective is to predict the future values for $H$ timestamps $(x_{t+1}, \ldots, x_{t+H})$. We use a similar model architecture to Pathformer, introduced by Chen et al. (2024), as our base model and extend it to leverage unseen data. Our model architecture for long-term time series forecasting is depicted in Figure 1.

To effectively capture multi-scale characteristics, we employ multi-scale patch division with various patch sizes $(P_1, P_2, \ldots, P_m)$ and dual attention transformer. The multi-scale patch division provides different views of the time series at different resolutions. These patches are fed into the transformer encoder, which comprises intra-patch attention within each divided patch and inter-patch attention across different patches. Following the transformer encoder, Multi-Layer Perceptrons (MLPs) generate individual predictions $(\widehat{x}_i)$ for each patch size. These predictions are then aggregated using an equal weighted aggregation strategy, resulting in a single comprehensive prediction $(\widehat{x}_*)$. Unlike Chen et al. (2024), we use equal weighted aggregator to combine these multi-scale characteristics.

During supervised learning, we train the model to minimize the mean squared error (MSE) between the predictions and the ground truth. Given that there are $m + 1$ predictions, we average the errors of each prediction. The supervised loss is optimized as follows:

$$\mathcal{L}_s = \frac{1}{m+1} \left( ||\widehat{x_*}_{t+1:t+H} - x_{t+1:t+H}||^2 + \sum_{i=1}^{m} ||\widehat{x_i}_{t+1:t+H} - x_{t+1:t+H}||^2 \right)$$

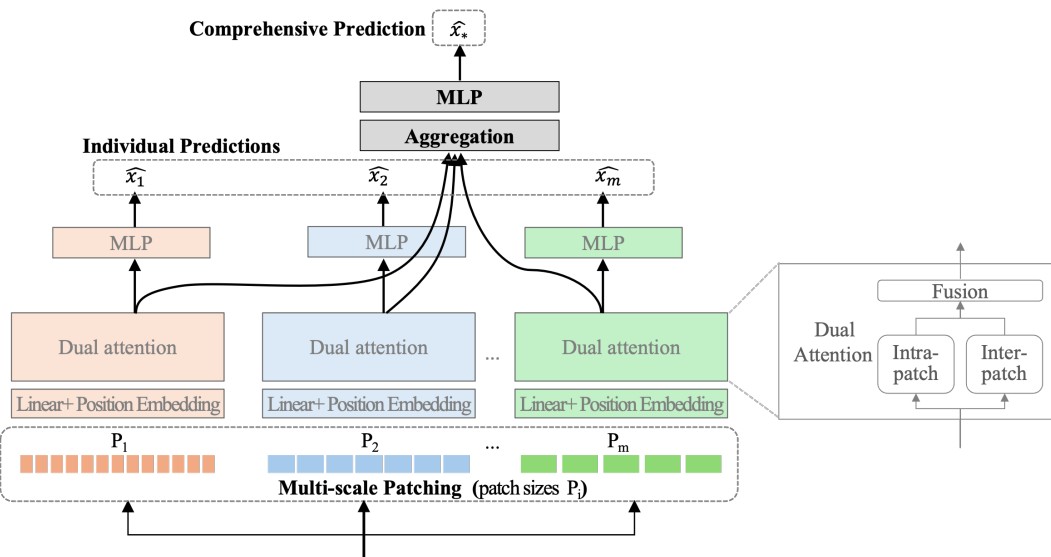

Figure 1: Overall architecture of multi-resolution transformer, CoMRes and MRes. There are $m$ individual blocks and a aggregation block. Each individual block predict *individual* prediction, $\widehat{x}_i$. The aggregation block produce *comprehensive* prediction, $\widehat{x}_*$.

In the following sections, we provide a detailed description of consensus promotion learning on unseen data and the process of generating unseen data. Subsequently, we explain how to extend prediction horizons beyond the training ranges.

## 3.1 CONSENSUS PROMOTION LEARNING

Due to the complex dynamics of long-term time series data, learning models from diverse and unseen data is crucial for improving forecasting accuracy. However, determining the future values of unseen augmented data is challenging, as this data is synthetic and potentially noisy. To address this issue, we propose a method inspired by standard multi-view learning approaches, which aim to jointly optimize multiple models derived from distinct perspectives, each providing complementary information. Drawing from the concept of one model teaching others, as introduced by Blum & Mitchell (1998) and Lim et al. (2020), we suggest using a *consistency loss* that encourages mutual agreement among the models. This approach leverages the unique information provided by different views to enhance comprehensive predictions ($\widehat{x}_*$), effectively capturing the complex, multi-resolution patterns inherent in long-term time series forecasting.

To achieve this, we propose that each individual-view model learns from the comprehensive prediction ($\widehat{x}_*$) by optimizing the following unsupervised loss:

$$\mathcal{L}_u = \frac{1}{m} \left( \sum_{i=1}^{m} ||\widehat{x}_{i\,t+1:t+H} - \widehat{x}_{*\,t+1:t+H}||^2 \right)$$

This encourages the individual-view models to align their predictions with the comprehensive prediction, thereby enhancing mutual agreement among the models. The overall objective for training the model combines the supervised loss and the unsupervised consistency loss as follows:

$$\mathcal{L} = \mathcal{L}_s + w_u \mathcal{L}_u$$

where $w_u$ represents the weight ot the unsupervised consistency loss $\mathcal{L}_u$. This joint objective ensures that the model not only minimizes the prediction error with respect to the ground truth($\mathcal{L}_s$) but also aligns the individual model predictions with the comprehensive prediction($\mathcal{L}_u$).

The core principle of consensus promotion learning is to achieve best possible forecasting results on unseen data. By leveraging the complementary information from multiple views and aligning individual model predictions with a comprehensive prediction, it reduces the variance of individual

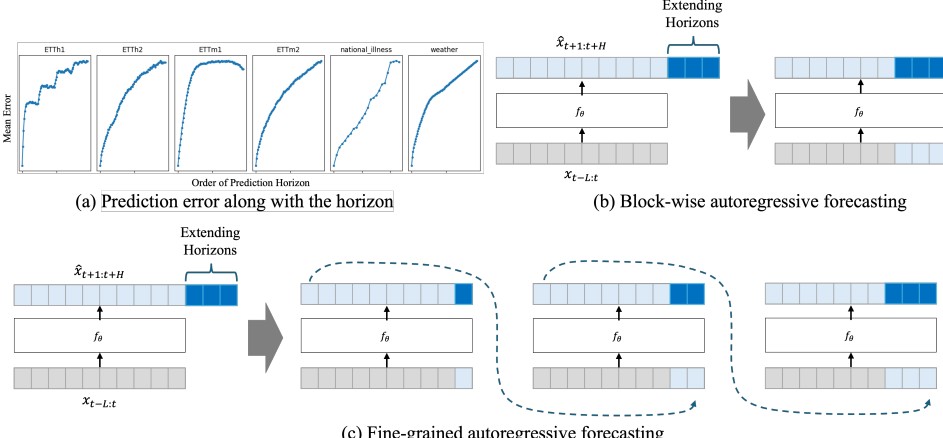

Figure 2: Extending Prediction Horizons Beyond Training Ranges. Figure (a) shows error rising with longer horizons. Figure (b) uses block-wise forecasting for the entire horizon, while Figure (c) adopts a step-by-step fine-grained approach. Both methods apply autoregression with a non-autoregressive trained model, but the fine-grained approach utilizes the initial prediction due to its relatively small error.

predictions(see Appendix A.2), leading to more stable and reliable results on unseen data. This is because the consensus or ensemble of predictions can help mitigate the biases and errors that may be present in any single model, allowing for better generalization to new, previously unobserved patterns.

## 3.2 UNSEEN DATA GENERATION

We generate unseen data via augmentations (time warping, interpolation, noise injection). The objective of generating unseen data is to construct a synthetic data that explores new areas of the input space. In supervised learning, data augmentation typically assumes that the relationship between the lookback period and the prediction horizon is well-defined, with known ground-truth labels. In contrast, our approach does not require defining ground truth for augmented data, offering greater flexibility for complex time series.

Time series data augmentation for forecasting has been extensively studied (Wen et al., 2021b; Debnath et al., 2021; Iglesias et al., 2023; Semenoglou et al., 2023; Zhou et al., 2023; Nochumsohn & Azencot, 2025), but we focus on simpler techniques for their ease of implementation. We include three variants of CoMRes: interpolation, time warping, and noise injection. Combining multiple augmentation techniques to enhance model performance will be discussed in Section 4.3.2.

**Interpolation** involves estimating values between known data points to produce a smoother and more continuous time series. For example, linear interpolation estimates values between two known points using a straight line. In this study, we adopt linear interpolation.

**Time Warping** includes techniques such as stretching or compressing the time window of data. Similar to dynamic time warping (DTW), this method selects a random time range and then compresses (down-samples) it. In this study, we compress the time window by a factor of 3.

**Noise Injection** adds random noise to the data to introduce variability. This includes injecting Gaussian noise, spike, step-like trend, and slope-like trend, etc. In this study, Gaussian noise, which consists of random values drawn from a normal distribution, is used. After normalization, we add normal noise with a mean of 0 and a standard deviation of 0.01

## 3.3 EXTENDING PREDICTION HORIZONS BEYOND TRAINING RANGES

In addition to the typical evaluation of forecasting performance, which involves measuring the prediction error over $H$ timestamps (training horizon), we also assess our model's performance when

predicting beyond this training horizon. For applications requiring long-term future predictions, such as stock prices or energy consumption, autoregressive forecasting is essential. However, autoregressive forecasting inherently suffers from error accumulation, especially when the model has not been trained autoregressively over an extended horizon, such as with teacher forcing. As depicted in Figure 2(a), our non-autoregressive model's prediction errors escalate as the prediction horizon lengthens. Our objective is to explore methods that can extend the prediction horizon while minimizing error accumulation in autoregressive forecasting scenarios.

In this study, we evaluate the model trained with consensus promotion through two approaches: Block-wise autoregressive prediction and fine-grained autoregressive prediction. The block-wise autoregressive prediction method, illustrated in Figure 2(b), extends the prediction horizon by utilizing the entire prediction range at once. In contrast, fine-grained autoregressive prediction involves a step-by-step process where the output of one step serves as the input for the next step, emphasizing a recurrent or sequential processing approach, as shown in Figure 2(c). This progression demonstrates the iterative application of the model to refine predictions over multiple steps.

## 4 EXPERIMENTS

### 4.1 TIME SERIES FORECASTING

**Datasets.** We evaluate our model on 8 popular datasets, including 4 ETT datasets (ETTh1, ETTh2, ETTm1, ETTm2), Weather, Traffic, Electricity, ILI. These datasets have been extensively utilized for benchmarking and publicly available on Wu et al. (2021) and the data statistics are shown in Appendix A.1.1.

**Baseline.** We compare CoMRes with Pathformer Chen et al. (2024), one of the state-of-the-art long-term forecasting models, which uses a multi-resolution division architecture with adaptive pathways. Pathformer outperforms most other models, such as TimeMixer, NLinear, and others (as shown in Appendix Table 8). Additionally, results for another baseline model, TimeMixer, are provided in Appendix A.5 to demonstrate the general application of CoMRes.

**Ablations.** To evaluate the impact of two core strategies—consensus promotion and data augmentation—on both labeled and unlabeled data, our proposed model (CoMRes) and its ablation models (MRes, short for **M**ulti-**Res**olution aggregation transformer) share the same architecture, as illustrated in Figure 1. We established the following three ablation models:

- **MRes (SL)**: MRes trained exclusively on labeled datasets using supervised learning (SL).
- **MRes w. augmentation**: MRes trained on pseudo-labeled datasets augmented with Time Warping.
- **MRes w. consensus**: MRes trained on labeled datasets with consensus promotion but without data augmentation. Predictions on labeled data are learned not only from the ground truth but also from the comprehensive prediction.

MRes with both data augmentation and consensus promotion applied to all augmented data corresponds to CoMRes.

**Experimental Setup.** To assess the impact of different unseen data generation methods, we include three variants of CoMRes: interpolation, time warping, and noise injection. Each dataset was split into training, validation, and test sets. Mean squared error (MSE) and mean absolute error (MAE) were used as evaluation metrics, with each experiment repeated five times, and the average values reported (see Appendix A.4 for error bars). Following the setup from the Pathformer paper, all models used the same input and prediction lengths: $L = 36$ for the ILI dataset and $L = 96$ for the others. Prediction lengths were $H \in \{24, 36, 48, 60\}$ for ILI and $H \in \{96, 192, 336, 720\}$ for the others. More experimental details are in Appendix A.1.

**Results.** The experimental results, summarized in Table 1, indicate that our proposed model outperforms the baselines on most datasets. Notably, CoMRes, which incorporates consensus promotion on unseen data significantly enhanced the model's generalization ability, as evidenced by lower MSE scores on the test sets. Although MRes with consensus promotion showed good results in some datasets, these improvements were inconsistent across others. Both CoMRes and MRes with

Table 1: Multivariate time series forecasting results (MSE). The input length $L = 96$ ($L = 36$ for the ILI dataset). The best results are highlighted in **bold**, and the second-best results are underlined. The $\Delta$ column shows the difference between CoMRes's best results and the ablation models or the baseline, with blue indicating that CoMRes outperforms and red indicating otherwise.

| Method | | CoMRes (ours) | | | MRes | | | | | | Pathformer | $\Delta$ |
|---|---|---|---|---|---|---|---|---|---|---|---|---|
| | | Time Warping | Interpolation | Noise Injection | w. augmentation | $\Delta$ | w. consensus | $\Delta$ | MRes (SL) | $\Delta$ | | |
| Consensus Promotion | | ○ | ○ | ○ | × | | ○ | | × | | | |
| Leveraging augmented Data | | ○ | ○ | ○ | ○ (pseudo label) | | × | | × | | | |
| ETTh1 | 96 | **0.378** | 0.379 | 0.379 | 0.406 | 0.028 | 0.379 | 0.001 | 0.382 | 0.004 | 0.382 | 0.004 |
| | 192 | **0.436** | **0.436** | **0.436** | 0.455 | 0.019 | **0.436** | 0.000 | 0.439 | 0.003 | 0.440 | 0.004 |
| | 336 | 0.452 | 0.452 | 0.452 | 0.482 | 0.030 | 0.452 | 0.000 | 0.451 | -0.001 | 0.454 | 0.002 |
| | 720 | **0.474** | 0.478 | 0.478 | 0.475 | 0.001 | 0.478 | 0.004 | 0.481 | 0.007 | 0.479 | 0.005 |
| | Avg. | **0.435** | 0.436 | 0.436 | 0.455 | 0.020 | 0.436 | 0.001 | 0.438 | 0.003 | 0.439 | 0.004 |
| ETTh2 | 96 | **0.275** | **0.275** | **0.275** | 0.278 | 0.003 | **0.275** | 0.000 | 0.277 | 0.002 | 0.279 | 0.004 |
| | 192 | 0.347 | **0.346** | **0.346** | 0.355 | 0.009 | **0.346** | 0.000 | 0.348 | 0.002 | 0.349 | 0.003 |
| | 336 | **0.326** | 0.328 | **0.326** | 0.346 | 0.020 | 0.330 | 0.004 | 0.332 | 0.006 | 0.348 | 0.022 |
| | 720 | **0.393** | 0.398 | 0.397 | 0.404 | 0.011 | 0.405 | 0.012 | 0.406 | 0.013 | 0.398 | 0.005 |
| | Avg. | **0.335** | 0.337 | 0.336 | 0.346 | 0.011 | 0.339 | 0.004 | 0.341 | 0.006 | 0.344 | 0.009 |
| ETTm1 | 96 | **0.311** | **0.311** | **0.311** | 0.326 | 0.016 | **0.311** | 0.001 | 0.312 | 0.001 | 0.316 | 0.005 |
| | 192 | 0.362 | 0.364 | 0.364 | 0.367 | 0.005 | 0.364 | 0.002 | 0.361 | -0.001 | 0.366 | 0.004 |
| | 336 | 0.386 | 0.386 | 0.387 | 0.404 | 0.018 | 0.386 | 0.000 | 0.381 | -0.005 | 0.386 | 0.000 |
| | 720 | 0.456 | 0.456 | 0.456 | 0.467 | 0.011 | 0.456 | 0.000 | 0.454 | -0.002 | 0.460 | 0.004 |
| | Avg. | 0.379 | 0.379 | 0.380 | 0.391 | 0.013 | 0.379 | 0.001 | 0.377 | -0.002 | 0.382 | 0.003 |
| ETTm2 | 96 | **0.163** | **0.163** | **0.163** | 0.168 | 0.005 | 0.166 | 0.003 | 0.167 | 0.004 | 0.170 | 0.007 |
| | 192 | **0.229** | **0.229** | **0.229** | 0.230 | 0.001 | **0.229** | 0.000 | 0.231 | 0.002 | 0.238 | 0.009 |
| | 336 | **0.291** | 0.292 | 0.292 | 0.295 | 0.004 | 0.292 | 0.001 | 0.292 | 0.001 | 0.293 | 0.002 |
| | 720 | **0.369** | **0.369** | **0.369** | 0.385 | 0.016 | 0.383 | 0.014 | **0.369** | 0.000 | 0.390 | 0.021 |
| | Avg. | **0.263** | **0.263** | **0.263** | 0.270 | 0.007 | 0.268 | 0.005 | 0.265 | 0.002 | 0.273 | 0.010 |
| Electricity | 96 | 0.152 | 0.153 | 0.153 | 0.162 | 0.010 | 0.156 | 0.004 | 0.150 | -0.002 | **0.145** | 0.007 |
| | 192 | 0.167 | 0.167 | 0.166 | 0.173 | 0.007 | 0.169 | 0.002 | 0.164 | -0.002 | 0.167 | 0.001 |
| | 336 | **0.181** | **0.181** | **0.181** | 0.186 | 0.005 | 0.184 | 0.003 | **0.181** | 0.000 | 0.186 | 0.005 |
| | 720 | **0.210** | 0.211 | 0.214 | 0.218 | 0.008 | 0.214 | 0.004 | 0.223 | 0.013 | 0.231 | 0.021 |
| | Avg. | **0.178** | **0.178** | 0.179 | 0.185 | 0.007 | 0.181 | 0.003 | 0.180 | 0.002 | 0.182 | 0.004 |
| ILI | 24 | 1.957 | 1.907 | 1.956 | 1.760 | -0.147 | 1.986 | 0.029 | 1.976 | 0.069 | **1.587** | -0.320 |
| | 36 | 1.640 | 1.599 | 1.538 | 1.569 | 0.031 | 1.540 | -0.100 | 1.551 | 0.013 | **1.429** | -0.109 |
| | 48 | 1.529 | 1.516 | 1.490 | 1.514 | 0.024 | **1.456** | -0.073 | 1.497 | 0.007 | 1.505 | 0.015 |
| | 60 | 1.693 | 1.739 | 1.715 | 1.727 | 0.034 | **1.679** | -0.014 | 1.714 | 0.021 | 1.731 | 0.038 |
| | Avg. | 1.705 | 1.690 | 1.675 | 1.643 | -0.032 | 1.665 | -0.039 | 1.685 | 0.010 | **1.563** | 0.112 |
| Traffic | 96 | **0.468** | 0.470 | 0.470 | 0.475 | 0.007 | 0.479 | 0.011 | 0.478 | 0.010 | 0.479 | 0.011 |
| | 192 | **0.465** | 0.466 | 0.466 | 0.482 | 0.017 | 0.479 | 0.014 | 0.476 | 0.011 | 0.484 | 0.019 |
| | 336 | **0.493** | 0.494 | 0.494 | 0.511 | 0.018 | 0.501 | 0.008 | 0.501 | 0.008 | 0.503 | 0.010 |
| | 720 | **0.535** | **0.535** | **0.535** | 0.540 | 0.005 | 0.537 | 0.002 | 0.556 | 0.021 | 0.537 | 0.002 |
| | Avg. | **0.490** | 0.491 | 0.491 | 0.502 | 0.012 | 0.499 | 0.009 | 0.503 | 0.013 | 0.501 | 0.011 |
| Weather | 96 | **0.151** | **0.151** | **0.151** | 0.155 | 0.004 | 0.154 | 0.003 | 0.152 | 0.001 | 0.156 | 0.005 |
| | 192 | **0.199** | 0.200 | 0.200 | 0.203 | 0.004 | 0.201 | 0.002 | **0.199** | 0.000 | 0.206 | 0.007 |
| | 336 | **0.244** | 0.246 | 0.246 | 0.248 | 0.010 | 0.246 | 0.004 | 0.245 | 0.002 | 0.254 | 0.010 |
| | 720 | 0.335 | 0.335 | 0.334 | 0.334 | 0.000 | **0.333** | -0.002 | 0.334 | 0.000 | 0.340 | 0.006 |
| | Avg. | **0.232** | 0.233 | 0.233 | 0.235 | 0.003 | 0.234 | 0.001 | 0.233 | 0.001 | 0.239 | 0.007 |

consensus incorporate consensus promotion; however, CoMRes applies it to unseen data, whereas MRes with consensus applies it to labeled data. This distinction is critical, as adding a consistency loss term alongside supervised loss can improve performance but also carries the risk of conflicting signals when applied to labeled data. Additionally, MRes with augmentation where augmented data is artificial and potentially noisy data performs poorly. Our results indicate that CoMRes consistently outperforms the baselines across multiple datasets. The case of ILI, where CoMRes underperforms compared to Pathformer, is discussed in detail in Appendix A.9. Interestingly, MRes (SL) outperformed Pathformer in most cases, suggesting that Pathformer's strengths lie in its ability to process multi-resolution information and its inter- and intra-attention design, rather than in the use of adaptive pathways. Among the unseen data generation techniques of CoMRes, time warping achieved slightly better performance compared to interpolation and noise injection. Overall, these findings demonstrate the efficacy of our approach in improving time series forecasting performance. Complementary results for mean absolute error(MAE) are provided in the Appendix A.3.

Figure 3 illustrates the mean error against the prediction horizon, demonstrating how prediction errors increase as the forecast extends further into the future from the lookback period. As shown in Figure 3, our proposed model consistently outperforms the baseline model in terms of prediction accuracy, especially over longer horizons. This comparison underscores the effectiveness of our approach in reducing errors in long-term time series forecasting.

## 4.2 ERROR ACCUMULATION IN AUTO-REGRESSIVE FORECASTING

**Experimental Setting.** In this section, we conduct experiments to evaluate our proposed model (CoMRes, Time Warping) and the ablation model (MRes, SL) on test data, focusing on predictions beyond the training horizon. The datasets used are described in Section 4.1.

We employ two approaches for autoregressive prediction. In both approaches, the extended prediction horizon is 96 sequences beyond the model's initial prediction horizon.

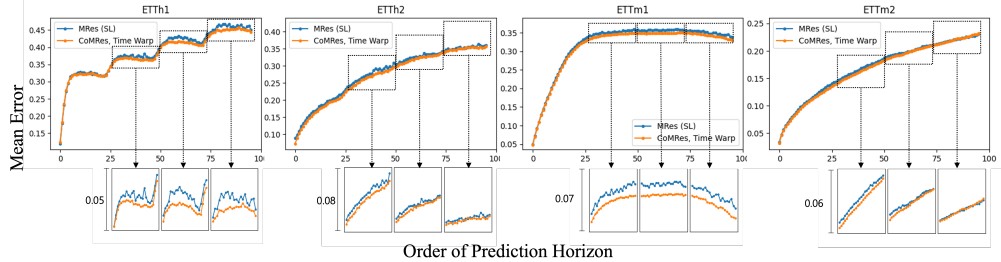

Figure 3: Prediction error along with the prediction horizon. The orange line representing the mean error of the proposed method is lower than that of MRes (SL), the blue line.

Table 2: Auto-regressive forecasting result. MSE on the extended horizon beyond the trained horizon. The best results are highlighted in **bold**. In both (a) and (b), CoMRes predicts more accurately than MRes. Additionally, (a) block-wise approach produces more accurate predictions compared to (b) fine-grained approach.

| Method | | (a) Block-wise Auto-regressive | | (b) Fine-grained Auto-regressive | |
|---|---|---|---|---|---|
| Data | Horizon | CoMRes, Time Warp | MRes (SL) | CoMRes, Time Warp | MRes (SL) |
| ETTh1 | 96 + 96 | **0.465** | 0.477 | 0.469 | 0.490 |
| | 192 + 96 | **0.476** | 0.490 | 0.481 | 0.499 |
| | 336 + 96 | **0.473** | 0.491 | 0.481 | 0.506 |
| | 720 + 96 | 0.505 | 0.513 | **0.500** | 0.507 |
| ETTh2 | 96 + 96 | **0.420** | 0.453 | 0.429 | 0.465 |
| | 192 + 96 | **0.428** | 0.451 | 0.430 | 0.455 |
| | 336 + 96 | 0.430 | 0.463 | **0.429** | 0.468 |
| | 720 + 96 | 0.445 | **0.437** | 0.448 | 0.469 |
| ETTm1 | 96 + 96 | **0.412** | 0.414 | 0.424 | 0.442 |
| | 192 + 96 | **0.416** | 0.418 | 0.421 | 0.429 |
| | 336 + 96 | **0.426** | 0.429 | 0.430 | 0.432 |
| | 720 + 96 | **0.449** | 0.451 | 0.454 | 0.455 |
| ETTm2 | 96 + 96 | **0.332** | 0.336 | 0.351 | 0.416 |
| | 192 + 96 | 0.358 | **0.355** | 0.361 | 0.404 |
| | 336 + 96 | **0.374** | 0.377 | 0.384 | 0.378 |
| | 720 + 96 | **0.417** | 0.420 | 0.420 | 0.421 |
| Weather | 96 + 96 | **0.280** | 0.285 | 0.303 | 0.425 |
| | 192 + 96 | **0.303** | 0.305 | 0.303 | 0.425 |
| | 336 + 96 | **0.323** | 0.347 | 0.331 | 0.356 |
| | 720 + 96 | **0.358** | 0.361 | 0.361 | 0.372 |

- **Block-wise Autoregressive Prediction**: The model generates predictions for the entire extended horizon in one step. Specifically, the model uses part of its own prediction as input to produce predictions beyond this range at once.

- **Fine-grained Autoregressive Prediction**: The model generates predictions step-by-step, where each output serves as the input for the next prediction step. As depicted in Figure 2(a), the initial prediction error over the prediction horizon is relatively small. Therefore, if we use the initial prediction as an input to extend the horizon step by step, we can expect to refine the predictions over multiple steps. In this approach, we update the input with the last $k=1$ value using the previous prediction.

**Auto-regressive Forecasting Results.** Table 2 presents the results of our auto-regressive forecasting evaluation. Compared to the errors shown in Table 1, the mean errors increased due to error accumulation. This issue is inherent in auto-regressive forecasting, particularly when the model has not been trained auto-regressively over an extended horizon, such as with teacher forcing. Despite these challenges, CoMRes consistently outperformed the MRes(SL) across all prediction horizons and both auto-regressive methods, demonstrating superior predictive accuracy. The block-wise auto-regressive method generally shows smaller error accumulation compared to the fine-grained auto-regressive method. This suggests that the block-wise approach handles auto-regressive error accumulation better.

To investigate the accumulation of errors in block-wise autoregressive forecasting, we compared the prediction error over the entire horizon between a model trained on a longer forecast range and

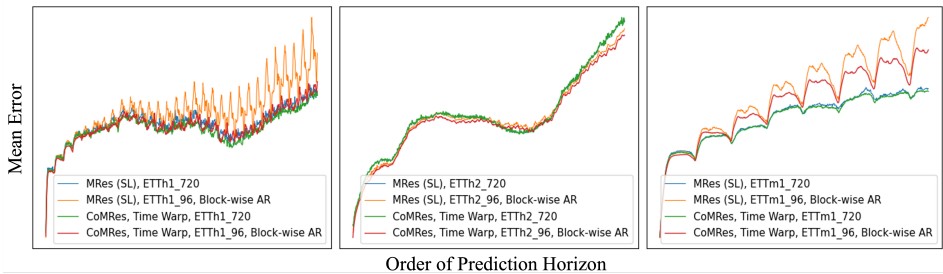

Figure 4: Error Propagation. The red line(CoMRes, Time Warp, 96 block-wise AR) shows less error accumulation in long-term forecasting compared to the orange line(MRes(SL), 96 block-wise AR), indicating that the our proposed method is more effective at mitigating error accumulation.

one using block-wise autoregressive predictions made with a shorter model. Figure 4 illustrates the performance of block-wise autoregressive forecasting with a shorter predictor model, specifically one trained to predict 96 steps, which is iteratively applied 8 times to forecast longer horizons (up to 768 time points). This approach is compared against the ablation model trained to predict 720 steps directly.

In the block-wise autoregressive setup, MRes (represented by the orange lines in the 96-step block-wise autoregressive approach) suffers from a substantial increase in error as the prediction horizon extends, especially when iterating the shorter model. This greater error accumulation in the supervised learning makes it more susceptible to compounding small errors with each iteration. On the other hand, the CoMRes (shown by the green and red lines) demonstrates a stronger capacity to reduce error accumulation, maintaining significantly lower error rates even when using block-wise autoregressive forecasting. While error accumulation occurs in CoMRes, for some data, the block-wise autoregressive model's error is comparable to that of the longer horizon non-autoregressive model. This indicates that it is feasible to effectively extend the prediction horizon of CoMRes using a shorter model.

## 4.3 ABLATION STUDY

### 4.3.1 LIMITED-RESOURCE SCENARIOS

Real-world time series often exhibit complex temporal variations, making it crucial to effectively leverage training data under resource constraints. We evaluated limited-data scenarios by systematically reducing labeled training data (20%, 50%, 80%) and comparing our CoMRes Time Warp against MRes across multiple datasets.

The results in Table 3 demonstrate that CoMRes consistently achieves the best performance across all label sizes and prediction horizons compared to the baseline[1]. Due to temporal variations, the temporal dynamics in distant periods may differ from those in the test period, leading to irrelevant information being included and causing worse predictions even with a large amount of data. Furthermore, on the ETTh2 and ETTm2 datasets, CoMRes delivers comparable performance to MRes, even at smaller label sizes.

### 4.3.2 COMBINING MULTIPLE UNSEEN DATA GENERATION TECHNIQUES

In this section, we evaluate the impact of combining multiple unseen data generation techniques. We examine two approaches: (1) Combining all three augmentation techniques—Time Warp, Interpolation, and Noise Injection—and (2) applying Multi Time Warp, using compression factors of 3, 5, and 7. These methods are compared against the best results achieved by CoMRes in 4.1, primarily using Time Warp, which we refer to as Single Time Warp.

The results in Table 4 indicate that the effectiveness of augmentation techniques depends on the dataset's characteristics. With high variability over time, such as ETTh1, combining multiple augmentation techniques did not lead to noticeable performance gains. However, with clear and consistent

---

[1]The Traffic results were obtained using H100 GPUs during the rebuttal period.

Table 3: Result in Limited-Resource scenarios. The best results among the compared methods are highlighted in **bold**, while the best result of the remaining methods is underlined. Label size is a fraction of the original training data (e.g., 0.5 uses only the most recent half).

| Method | | CoMRes, Time Warp | | | | MRes (SL) | | | |
|--------|-----|-------|-------|-------|-------|-------|-------|-------|-------|
| Label Size | | 0.2 | 0.5 | 0.8 | 1.0 | 0.2 | 0.5 | 0.8 | 1.0 |
| ETTh1 | 96 | 0.390 | **0.377** | 0.378 | 0.378 | 0.393 | 0.379 | 0.381 | 0.382 |
| | 192 | 0.444 | **0.435** | 0.437 | 0.436 | 0.448 | 0.436 | 0.438 | 0.439 |
| | 336 | 0.452 | 0.448 | **0.444** | 0.452 | 0.449 | 0.447 | 0.445 | 0.451 |
| | 720 | 0.504 | 0.489 | **0.469** | 0.474 | 0.474 | 0.481 | 0.470 | 0.481 |
| ETTh2 | 96 | 0.296 | 0.277 | 0.276 | **0.275** | 0.295 | 0.280 | 0.277 | 0.277 |
| | 192 | 0.377 | 0.356 | 0.348 | **0.345** | 0.377 | 0.363 | 0.349 | 0.347 |
| | 336 | 0.375 | 0.359 | 0.334 | **0.329** | 0.376 | 0.358 | 0.334 | 0.330 |
| | 720 | 0.452 | 0.438 | 0.434 | **0.399** | 0.460 | 0.434 | 0.416 | 0.409 |
| ETTm1 | 96 | 0.336 | **0.310** | 0.313 | **0.310** | 0.337 | 0.311 | 0.311 | 0.312 |
| | 192 | 0.376 | **0.357** | 0.361 | 0.362 | 0.377 | **0.357** | 0.361 | 0.361 |
| | 336 | 0.401 | **0.378** | 0.390 | 0.386 | 0.399 | **0.378** | 0.384 | 0.381 |
| | 720 | 0.471 | **0.446** | 0.467 | 0.456 | 0.471 | 0.447 | 0.454 | 0.454 |
| ETTm2 | 96 | 0.178 | 0.171 | 0.168 | **0.163** | 0.179 | 0.170 | 0.168 | 0.166 |
| | 192 | 0.248 | 0.234 | 0.231 | **0.229** | 0.248 | 0.234 | 0.232 | **0.229** |
| | 336 | 0.313 | 0.299 | 0.295 | **0.291** | 0.312 | 0.299 | 0.294 | 0.292 |
| | 720 | 0.409 | 0.400 | 0.386 | **0.383** | 0.408 | 0.401 | 0.386 | **0.383** |
| Weather | 96 | 0.164 | 0.160 | 0.155 | **0.151** | 0.175 | 0.163 | 0.156 | 0.152 |
| | 192 | 0.213 | 0.223 | 0.204 | **0.199** | 0.223 | 0.210 | 0.204 | **0.199** |
| | 336 | 0.265 | 0.257 | 0.250 | **0.244** | 0.258 | 0.257 | 0.250 | 0.245 |
| | 720 | 0.351 | 0.342 | 0.338 | 0.335 | 0.343 | 0.343 | 0.338 | **0.334** |
| Traffic | 96 | 0.536 | 0.486 | **0.446** | 0.468 | 0.583 | 0.501 | 0.470 | 0.478 |
| | 192 | 0.512 | 0.492 | 0.536 | **0.465** | 0.529 | 0.484 | 0.562 | 0.476 |
| | 336 | 0.566 | 0.501 | 0.526 | **0.493** | 0.550 | 0.503 | 0.589 | 0.501 |
| | 720 | 0.578 | 0.539 | 0.548 | **0.535** | 0.597 | 0.540 | 0.561 | 0.556 |

periodic patterns over time, such as ETTm2, improvements were observed at certain prediction horizons, although rapid long-term trend shift caused performance declines at the 720-horizon. In the Weather dataset, which features inter-variable correlations, combining augmentation techniques resulted in performance gains. Overall, while multi-augmentation approaches do not universally improve performance, they can be beneficial for some datasets. The degree of improvement, varies according to the specific features of the dataset and the forecasting task at hand.

Table 4: Results of combining unseen data generation techniques. Performance improvements compared to the Single Time Warp method are in blue, while declines are in red.

| Method | | Combine all three | Multi Time Warp | Single Time Warp |
|--------|-----|-------------------|-----------------|------------------|
| ETTh1 | 96 | 0.378 | 0.379 | 0.378 |
| | 192 | 0.436 | 0.436 | 0.436 |
| | 336 | 0.452 | 0.452 | 0.452 |
| | 720 | 0.474 | 0.476 | 0.474 |
| ETTm2 | 96 | 0.164 | 0.163 | 0.163 |
| | 192 | 0.228 | 0.227 | 0.229 |
| | 336 | 0.289 | 0.288 | 0.291 |
| | 720 | 0.384 | 0.382 | 0.369 |
| Weather | 96 | 0.149 | 0.150 | 0.151 |
| | 192 | 0.198 | 0.199 | 0.199 |
| | 336 | 0.244 | 0.244 | 0.244 |
| | 720 | 0.330 | 0.334 | 0.334 |

## 5 CONCLUSION

This paper introduces a multi-view learning strategy for long-term time series forecasting, demonstrating superior prediction accuracy and robustness compared to traditional supervised learning approaches. This approach leverages augmented unseen data to ensure that models not only capture multi-resolution patterns but also provide consistent predictions when faced with new, unseen data. Our comprehensive experiments highlight the model's ability to generalize well across various datasets and handle challenging forecasting scenarios effectively, especially in autoregressive setups, where it mitigates the issue of error accumulation.

ACKNOWLEDGMENTS

This work was supported in part by the National Research Foundation of Korea (NRF) grant (RS-2023-00280883, RS-2023-00222663) and partially supported by New Faculty Startup Fund from Seoul National University

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

# A APPENDIX

## A.1 EXPERIMENTAL DETAILS

### A.1.1 DATASETS

We use eight popular datasets from Wu et al. (2021) for forecasting. The ETT dataset[2] includes data from two electricity transformers at two stations, capturing metrics such as load and oil temperature. This dataset is recorded at both 15-minute and 1-hour intervals, and is labeled as ETTh1, ETTh2,

---

[2]https://github.com/zhouhaoyi/ETDataset

ETTm1, and ETTm2. The Electricity dataset[3] contains hourly electricity consumption data from 321 users. The ILI dataset[4] collects weekly reports of patients with influenza-like illness, spanning 2002 to 2021, from the Centers for Disease Control and Prevention of the United States. The Traffic dataset[5] records road occupancy rates from various sensors on San Francisco freeways. Finally, the Weather dataset[6] comprises 21 meteorological indicators collected every 10 minutes in Germany.

Table 5: Statistics of popular datasets for benchmark.

| Datasets | ETTh1 | ETTh2 | ETTm1 | ETTm2 | Electricity | ILI | Traffic | Weather |
|---|---|---|---|---|---|---|---|---|
| Features | 7 | 7 | 7 | 7 | 321 | 7 | 862 | 21 |
| Timesteps | 17,420 | 17,420 | 69,680 | 69,680 | 26,304 | 966 | 17,544 | 52,696 |
| Split Ratio | 6:2:2 | 6:2:2 | 6:2:2 | 6:2:2 | 7:1:2 | 7:1:2 | 7:1:2 | 7:1:2 |

### A.1.2 DETAILS OF IMPLEMENTATION

To ensure fair comparisons, we use the same hyperparameters as Pathformer in Chen et al. (2024). The model is trained using the Adam optimizer (Kingma, 2014) with a learning rate of $10^{-3}$. The default loss function is L1 Loss, and early stopping is applied after 10 epochs if no improvement is observed. All experiments are implemented in PyTorch and executed on a NVIDIA A6000 48GB GPU.

The baseline model, MRes, includes four different patch sizes, same to those used in Pathformer repository[7]. These patch sizes are selected from the following options: $\{2, 3, 6, 12, 16, 24, 32\}$. Unlike Pathformer, which is composed of three multi-scale blocks, MRes does not utilize a hierarchical architecture. This allows us to argue that MRes achieves comparable predictive performance with fewer model parameters, offering a more efficient design.

CoMRes and MRes atgenerates one comprehensive prediction and $m$ individual predictions, so for the final model performance evaluation, $m + 1$ predictions were used in an ensemble. First, the average value of the $m$ individual predictions was calculated, and then it was averaged with the comprehensive prediction to form the final prediction.

### A.2 VARIANCE OF INDIVIDUAL-VIEW MODEL

The Figure 5 provides clear evidence that applying consistency loss improves consistency between individual-view models compared to MRes (SL). MRes with consensus and CoMRes exhibits a consistently smoother and lower trend than MRes(SL), indicating reduced variability in predictions. This smoother trend suggests that individual models are more aligned and consistent with one another after applying the consistency loss, effectively achieving the intended effect of consensus promotion.

### A.3 TIME SERIES FORECASTING

In section 4.1, we reported the model performance using Mean Squared Error (MSE). Additionally, Table 6 presents the results based on Mean Absolute Error (MAE), providing a complementary evaluation of the model's predictive accuracy.

### A.4 ERROR BARS

In this paper, we repeat all the experiments five times. In Table 7, we report the standard deviation of our model and the baseline model. CoMRes exhibits smaller standard deviations compared to Pathformer [8], indicating less variability in its predictions.

---

[3]https://archive.ics.uci.edu/ml/datasets/ElectricityLoadDiagrams20112014

[4]https://gis.cdc.gov/grasp/fluview/fluportaldashboard.html

[5]https://pems.dot.ca.gov/

[6]https://www.bgc-jena.mpg.de/wetter/

[7]https://github.com/decisionintelligence/pathformer

[8]We were unable to reproduce Pathformer results on the ILI dataset. Therefore, we use the mean values of the metrics as reported in the original Pathformer paper and could not obtain the corresponding error bars.

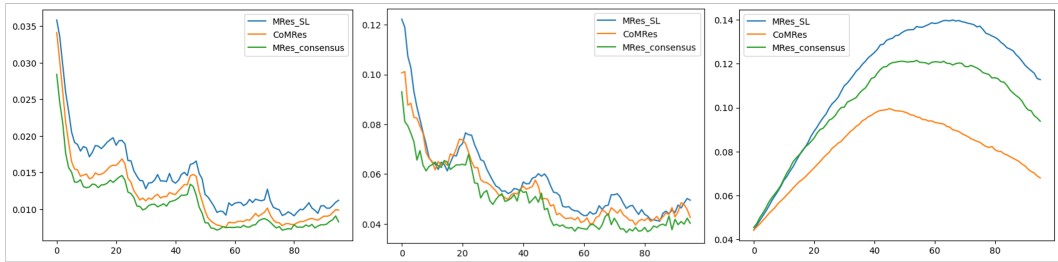

Figure 5: Comparison of individual-view model variability between MRes(SL), MRes with consensus and CoMRes across three datasets (ETTh1, ETTh2, and Weather) with a forecast horizon of h=96. MRes with consensus and CoMRes demonstrates consistently reduced variability compared to MRes(SL).

Table 6: Multivariate time series forecasting results (MAE). The input length L = 96 (L = 36 for ILI dataset). The best results are highlighted in bold, and the second-best results are underlined. The $\Delta$ column represents the difference between CoMRes best result and the baselines such as MRes with ablated components and Pathformer.

| Method | | CoMRes (ours) | | | MRes | | | | | | Pathformer | $\Delta$ |
|---|---|---|---|---|---|---|---|---|---|---|---|---|
| | | Time Warping | Interpolation | Noise Injection | w. augmentation | $\Delta$ | w. consensus | $\Delta$ | MRes (SL) | $\Delta$ | | |
| Consensus Promotion | | ○ | ○ | ○ | × | | ○ | | × | | | |
| Leveraging Augmented Data | | ○ | ○ | ○ | ○ (pseudo-labeled) | | × | | × | | | |
| ETTh1 | 96 | **0.386** | **0.386** | **0.386** | 0.415 | 0.029 | **0.386** | 0.000 | 0.387 | 0.001 | 0.400 | 0.014 |
| | 192 | **0.415** | **0.415** | **0.415** | 0.439 | 0.024 | **0.415** | 0.000 | 0.416 | 0.001 | 0.427 | 0.012 |
| | 336 | **0.421** | **0.421** | **0.421** | 0.458 | 0.037 | **0.421** | 0.000 | 0.421 | 0.000 | 0.432 | 0.011 |
| | 720 | 0.453 | 0.453 | 0.451 | 0.476 | 0.025 | 0.453 | 0.002 | 0.454 | 0.003 | 0.461 | 0.010 |
| | Avg. | 0.419 | 0.419 | 0.418 | 0.447 | 0.029 | 0.419 | 0.001 | 0.420 | 0.002 | 0.430 | 0.012 |
| ETTh2 | 96 | **0.327** | **0.327** | **0.327** | 0.332 | 0.005 | **0.327** | 0.000 | 0.329 | 0.002 | 0.331 | 0.004 |
| | 192 | **0.373** | 0.374 | **0.373** | 0.382 | 0.009 | 0.374 | 0.001 | 0.375 | 0.002 | 0.380 | 0.007 |
| | 336 | **0.369** | 0.370 | **0.369** | 0.385 | 0.016 | 0.371 | 0.002 | 0.373 | 0.004 | 0.382 | 0.013 |
| | 720 | 0.422 | 0.422 | 0.419 | 0.430 | 0.011 | 0.426 | 0.007 | 0.427 | 0.008 | 0.424 | 0.005 |
| | Avg. | 0.373 | 0.373 | **0.372** | 0.382 | 0.010 | 0.375 | 0.003 | 0.376 | 0.004 | 0.379 | 0.007 |
| ETTm1 | 96 | **0.339** | **0.339** | **0.339** | 0.360 | 0.021 | **0.339** | 0.000 | 0.341 | 0.002 | 0.346 | 0.007 |
| | 192 | **0.365** | **0.365** | **0.365** | 0.383 | 0.018 | **0.365** | 0.000 | 0.366 | 0.001 | 0.370 | 0.005 |
| | 336 | **0.387** | **0.387** | **0.387** | 0.404 | 0.017 | **0.387** | 0.000 | 0.387 | 0.000 | 0.394 | 0.007 |
| | 720 | **0.424** | 0.425 | 0.425 | 0.439 | 0.015 | 0.425 | 0.001 | 0.426 | 0.002 | 0.432 | 0.008 |
| | Avg. | **0.379** | **0.379** | **0.379** | 0.397 | 0.018 | **0.379** | 0.000 | 0.380 | 0.001 | 0.386 | 0.007 |
| ETTm2 | 96 | **0.245** | 0.246 | 0.246 | 0.250 | 0.005 | 0.246 | 0.001 | 0.247 | 0.002 | 0.248 | 0.002 |
| | 192 | 0.290 | 0.291 | 0.290 | 0.292 | 0.002 | **0.289** | -0.001 | 0.290 | 0.000 | 0.295 | 0.005 |
| | 336 | **0.330** | **0.330** | **0.330** | 0.334 | 0.004 | **0.330** | 0.000 | 0.330 | 0.000 | 0.331 | 0.001 |
| | 720 | **0.378** | 0.378 | 0.378 | 0.389 | 0.011 | 0.385 | 0.007 | 0.378 | 0.000 | 0.389 | 0.011 |
| | Avg. | **0.311** | **0.311** | **0.311** | 0.316 | 0.006 | 0.313 | 0.002 | **0.311** | 0.000 | 0.316 | 0.005 |
| Electricity | 96 | 0.240 | 0.240 | 0.239 | 0.247 | 0.008 | 0.243 | 0.004 | 0.237 | -0.002 | **0.236** | -0.003 |
| | 192 | 0.252 | 0.252 | 0.252 | 0.258 | 0.006 | 0.254 | 0.002 | **0.250** | -0.002 | 0.256 | 0.004 |
| | 336 | **0.267** | **0.267** | **0.267** | 0.272 | 0.005 | 0.269 | 0.002 | 0.269 | 0.002 | 0.275 | 0.008 |
| | 720 | 0.293 | 0.293 | **0.292** | 0.299 | 0.007 | 0.294 | 0.002 | 0.302 | 0.010 | 0.309 | 0.017 |
| | Avg. | **0.263** | **0.263** | **0.263** | 0.269 | 0.006 | 0.265 | 0.002 | 0.265 | 0.002 | 0.269 | 0.007 |
| ILI | 24 | 0.830 | 0.823 | 0.834 | 0.799 | -0.024 | 0.837 | 0.014 | 0.840 | 0.010 | **0.758** | -0.065 |
| | 36 | 0.755 | 0.764 | 0.778 | 0.741 | -0.014 | 0.749 | -0.006 | 0.758 | 0.003 | **0.711** | -0.044 |
| | 48 | 0.766 | 0.769 | 0.767 | 0.749 | -0.017 | 0.744 | -0.022 | 0.761 | -0.005 | **0.742** | -0.024 |
| | 60 | 0.811 | 0.821 | 0.808 | 0.811 | 0.003 | 0.808 | 0.000 | 0.810 | 0.002 | **0.799** | -0.009 |
| | Avg. | 0.791 | 0.794 | 0.797 | 0.775 | -0.016 | 0.785 | -0.006 | 0.792 | 0.001 | **0.753** | -0.038 |
| Traffic | 96 | **0.270** | 0.271 | 0.271 | 0.275 | 0.005 | 0.275 | 0.005 | 0.273 | 0.003 | 0.283 | 0.013 |
| | 192 | **0.285** | 0.286 | 0.286 | 0.292 | 0.007 | 0.282 | -0.003 | 0.284 | -0.001 | 0.292 | 0.007 |
| | 336 | **0.295** | **0.295** | **0.295** | 0.300 | 0.005 | **0.295** | 0.000 | 0.296 | 0.001 | 0.299 | 0.004 |
| | 720 | 0.324 | 0.324 | 0.324 | 0.333 | 0.009 | 0.323 | -0.001 | 0.321 | -0.003 | 0.322 | -0.002 |
| | Avg. | **0.294** | **0.294** | **0.294** | 0.300 | 0.007 | 0.294 | 0.000 | **0.294** | 0.000 | 0.299 | 0.006 |
| Weather | 96 | **0.190** | **0.190** | **0.190** | 0.195 | 0.005 | 0.193 | 0.003 | **0.190** | 0.000 | 0.192 | 0.002 |
| | 192 | 0.237 | 0.237 | **0.236** | 0.239 | 0.002 | 0.237 | 0.000 | **0.236** | 0.000 | 0.240 | 0.004 |
| | 336 | 0.276 | 0.276 | **0.275** | 0.279 | 0.003 | 0.276 | 0.000 | 0.276 | 0.001 | 0.282 | 0.007 |
| | 720 | 0.331 | 0.332 | 0.332 | 0.333 | 0.002 | **0.330** | -0.001 | 0.331 | 0.000 | 0.336 | 0.005 |
| | Avg. | 0.259 | 0.259 | **0.258** | 0.262 | 0.003 | 0.259 | 0.001 | **0.258** | 0.000 | 0.263 | 0.005 |

## A.5 ADDITIONAL APPLICATION OF COMRES TO TIMEMIXER

To examine the general applicability of the proposed CoMRes, we explore the applicability of consensus promotion on augmented data to other architectures, particularly TimeMixer(Wang et al., 2024). Table 8 demonstrates that applying consensus promotion on augmented data enhances TimeMixer's forecasting accuracy in most cases. The best-performing CoMRes results (across different augmentations) are compared against the reproduced TimeMixer results in the $\Delta$ column. In the majority of datasets, TimeMixer + CoMRes achieves lower MSE than the reproduced TimeMixer, indicating improved forecasting performance. While CoMRes generally contributes to better results, its effectiveness varies depending on the dataset and forecasting horizon.

In our adaptation of TimeMixer, we treat downpooled series as individual views. TimeMixer derives its final prediction by summing single-scale predictions, whereas Pathformer generates its final

Table 7: We provide standard deviation for our method and the MRes baselines over 5 independent runs. The asterisk (*) denotes statistically significant differences at the 95% confidence level.

| Method | | CoMRes (ours) | | | MRes | | | Pathformer |
|---|---|---|---|---|---|---|---|---|
| | | Time Warping | Interpolation | Noise Injection | w. augmentation | w. consensus | MRes (SL) | |
| Consensus Promotion | | ○ | ○ | × | × | ○ | × | |
| Leveraging Augmented Data | | ○ | ○ | ○ | ○ (pseudo-labeled) | × | × | |
| ETTh1 | 96 | 0.378±0.001 | 0.379±0.001 | 0.379±0.001 | 0.406±0.006* | 0.379±0.001 | 0.382±0.001* | 0.382±0.002* |
| | 192 | 0.436±0.002 | 0.436±0.001 | 0.436±0.001 | 0.455±0.006* | 0.436±0.001 | 0.439±0.001 | 0.440±0.001* |
| | 336 | 0.452±0.001 | 0.452±0.001 | 0.452±0.001 | 0.482±0.005* | 0.452±0.001 | 0.451±0.001* | 0.454±0.002* |
| | 720 | 0.474±0.003 | 0.478±0.003 | 0.478±0.003 | 0.475±0.004 | 0.478±0.003* | 0.481±0.002* | 0.479±0.001* |
| ETTh2 | 96 | 0.275±0.002 | 0.275±0.002 | 0.275±0.002 | 0.278±0.005 | 0.275±0.004 | 0.277±0.003 | 0.279±0.005 |
| | 192 | 0.347±0.002 | 0.346±0.002 | 0.346±0.001 | 0.355±0.009 | 0.346±0.003 | 0.348±0.001 | 0.349±0.003 |
| | 336 | 0.326±0.001 | 0.328±0.005 | 0.326±0.001 | 0.346±0.006* | 0.330±0.004 | 0.332±0.008 | 0.348±0.003* |
| | 720 | 0.393±0.002 | 0.398±0.009 | 0.397±0.009 | 0.404±0.015 | 0.405±0.006* | 0.406±0.009* | 0.398±0.012* |
| ETTm1 | 96 | 0.310±0.001 | 0.311±0.001 | 0.311±0.001 | 0.326±0.002* | 0.311±0.001 | 0.312±0.001 | 0.316±0.002 |
| | 192 | 0.362±0.001 | 0.364±0.001 | 0.364±0.001 | 0.367±0.002* | 0.363±0.002 | 0.361±0.002 | 0.366±0.002 |
| | 336 | 0.386±0.001 | 0.386±0.001 | 0.387±0.001 | 0.404±0.004* | 0.386±0.001 | 0.381±0.001 | 0.386±0.003 |
| | 720 | 0.456±0.001 | 0.456±0.002 | 0.456±0.001 | 0.467±0.002* | 0.456±0.002 | 0.454±0.001* | 0.460±0.003 |
| ETTm2 | 96 | 0.163±0.000 | 0.163±0.000 | 0.163±0.000 | 0.168±0.001* | 0.166±0.001 | 0.167±0.001* | 0.170±0.001* |
| | 192 | 0.229±0.001 | 0.229±0.001 | 0.229±0.001 | 0.230±0.000 | 0.229±0.001 | 0.231±0.001 | 0.238±0.002* |
| | 336 | 0.291±0.001 | 0.292±0.001 | 0.292±0.001 | 0.295±0.001* | 0.292±0.002 | 0.292±0.001 | 0.293±0.003* |
| | 720 | 0.369±0.003 | 0.369±0.003 | 0.369±0.003 | 0.385±0.002* | 0.383±0.002* | 0.369±0.003 | 0.390±0.004* |
| Electricity | 96 | 0.152±0.004 | 0.153±0.003 | 0.153±0.003 | 0.162±0.003 | 0.156±0.003 | 0.150±0.003 | 0.145±0.006 |
| | 192 | 0.167±0.001 | 0.167±0.001 | 0.166±0.002 | 0.173±0.003 | 0.169±0.003 | 0.164±0.003 | 0.167±0.001* |
| | 336 | 0.181±0.009 | 0.181±0.007 | 0.181±0.007 | 0.186±0.006 | 0.184±0.006 | 0.181±0.006 | 0.186±0.001 |
| | 720 | 0.210±0.002 | 0.211±0.002 | 0.214±0.003 | 0.218±0.005 | 0.214±0.005 | 0.223±0.005 | 0.231±0.002 |
| ILI | 24 | 1.957±0.072 | 1.907±0.062 | 1.956±0.080 | 1.760±0.067* | 1.986±0.112 | 1.976±0.051 | 1.587 |
| | 36 | 1.640±0.036 | 1.599±0.076 | 1.538±0.124 | 1.569±0.079 | 1.540±0.055* | 1.551±0.137 | 1.429 |
| | 48 | 1.529±0.097 | 1.516±0.075 | 1.490±0.101 | 1.514±0.126 | 1.456±0.048 | 1.497±0.040 | 1.505 |
| | 60 | 1.693±0.091 | 1.739±0.064 | 1.715±0.049 | 1.727±0.047 | 1.679±0.092 | 1.714±0.048 | 1.731 |
| Traffic | 96 | 0.468±0.004 | 0.470±0.003 | 0.470±0.003 | 0.475±0.003 | 0.479±0.003 | 0.478±0.003 | 0.479±0.007 |
| | 192 | 0.465±0.001 | 0.466±0.001 | 0.466±0.002 | 0.482±0.003 | 0.479±0.003 | 0.476±0.003 | 0.484±0.004 |
| | 336 | 0.493±0.009 | 0.494±0.007 | 0.494±0.007 | 0.511±0.006 | 0.501±0.006 | 0.501±0.006 | 0.503±0.004 |
| | 720 | 0.535±0.002 | 0.535±0.002 | 0.535±0.003 | 0.540±0.005 | 0.537±0.005 | 0.556±0.005 | 0.537±0.004 |
| Weather | 96 | 0.151±0.000 | 0.151±0.000 | 0.151±0.000 | 0.155±0.001 | 0.154±0.000* | 0.152±0.001 | 0.156±0.002 |
| | 192 | 0.199±0.001 | 0.200±0.003 | 0.200±0.003 | 0.203±0.001 | 0.201±0.001 | 0.199±0.001 | 0.206±0.002 |
| | 192 | 0.244±0.001 | 0.246±0.002 | 0.246±0.002 | 0.248±0.002 | 0.246±0.000 | 0.245±0.001 | 0.254±0.002 |
| | 192 | 0.335±0.004 | 0.335±0.004 | 0.334±0.004 | 0.334±0.003 | 0.333±0.002 | 0.334±0.004 | 0.340±0.003 |

prediction through an additional projection layer after aggregating individual predictions. To facilitate complementary learning at the individual prediction level while avoiding conflicting ground truth signals, we apply the supervised loss only to TimeMixer's final prediction, rather than enforcing an ensemble-supervised loss on individual predictions.

The reported TimeMixer results could not be fully replicated with the hyperparameters provided in the official code (https://github.com/kwuking/TimeMixer). For a fair comparison, we performed a hyperparameter search to reproduce TimeMixer as a baseline and applied the same hyperparameter (e.g., learning rate, batch size) for expanding TimeMixer with CoMRes's consensus promotion on augmented data, maintaining consistency in experimental conditions. As our reporeduced results are slightly lower than the reported TimeMixer results, we also display the originally reported results in the last column a reference.

## A.6 Additional Baselines

To provide additional evaluation results and contextualize current progress in long-term time series forecasting, we include several advanced baselines in Table 9: Pathformer (Chen et al., 2024), PatchTST (Nie et al., 2023), TimeMixer (Wang et al., 2024), NLinear (Zeng et al., 2023), Scaleformer (Shabani et al., 2023), and TiDE (Das et al., 2023). All models use the same input length ($H = 36$ for the ILI dataset and $H = 96$ for others) and prediction lengths ($F \in \{24, 36, 48, 60\}$ for the ILI dataset and $F \in \{96, 192, 336, 720\}$ for others). We evaluate model performance using two common metrics in time series forecasting: Mean Absolute Error (MAE) and Mean Squared Error (MSE). The results for TimeMixer are taken from Wang et al. (2024), those for PatchTST are sourced from Nie et al. (2023), and the remaining results are obtained from Chen et al. (2024).

## A.7 Limited-Resource Scenarios

As noted in Section 4.3.1, CoMRes consistently outperforms MRes across all label sizes and prediction horizons in limited resource scenarios. Not only does CoMRes achieve the best scores, as presented

Table 8: Results of Applying Consensus Promotion on Augmented Data to TimeMixer. The reported TimeMixer results are taken from the original paper, while the reproduced results are obtained through our replication. The Δ column shows the difference between CoMRes's best results and TimeMixer, with blue indicating that CoMRes outperforms and red indicating otherwise.

| Method | | TimeMixer + CoMRes (ours) | | | TimeMixer | | | |
|---|---|---|---|---|---|---|---|---|
| | | Time Warping | Interpolation | Noise Injection | Reproduced | Δ | Reported | Δ |
| ETTh1 | 96 | 0.379 | 0.376 | 0.376 | 0.377 | 0.001 | 0.375 | -0.001 |
| | 192 | 0.436 | 0.431 | 0.430 | 0.431 | 0.001 | 0.429 | -0.001 |
| | 336 | 0.486 | 0.484 | 0.484 | 0.485 | 0.001 | 0.484 | 0.000 |
| | 720 | 0.499 | 0.489 | 0.489 | 0.498 | 0.009 | 0.498 | 0.009 |
| | Avg. | 0.450 | 0.445 | 0.445 | 0.448 | 0.003 | 0.447 | 0.002 |
| ETTh2 | 96 | 0.294 | 0.294 | 0.294 | 0.295 | 0.001 | 0.289 | -0.005 |
| | 192 | 0.374 | 0.374 | 0.374 | 0.375 | 0.001 | 0.372 | -0.002 |
| | 336 | 0.388 | 0.388 | 0.388 | 0.388 | 0.000 | 0.386 | -0.002 |
| | 720 | 0.419 | 0.419 | 0.419 | 0.422 | 0.003 | 0.412 | -0.007 |
| | Avg. | 0.369 | 0.369 | 0.369 | 0.370 | 0.001 | 0.365 | -0.004 |
| ETTm1 | 96 | 0.322 | 0.331 | 0.331 | 0.331 | 0.009 | 0.320 | -0.002 |
| | 192 | 0.364 | 0.364 | 0.364 | 0.369 | 0.006 | 0.361 | -0.003 |
| | 336 | 0.398 | 0.396 | 0.397 | 0.398 | 0.002 | 0.390 | -0.006 |
| | 720 | 0.465 | 0.458 | 0.460 | 0.461 | 0.004 | 0.454 | -0.004 |
| | Avg. | 0.387 | 0.387 | 0.388 | 0.390 | 0.003 | 0.381 | -0.006 |
| ETTm2 | 96 | 0.175 | 0.174 | 0.174 | 0.175 | 0.001 | 0.175 | 0.001 |
| | 192 | 0.239 | 0.240 | 0.240 | 0.242 | 0.003 | 0.237 | -0.002 |
| | 336 | 0.294 | 0.295 | 0.295 | 0.294 | 0.000 | 0.298 | 0.004 |
| | 720 | 0.390 | 0.389 | 0.389 | 0.393 | 0.003 | 0.391 | 0.002 |
| | Avg. | 0.274 | 0.275 | 0.275 | 0.276 | 0.002 | 0.275 | 0.001 |
| Weather | 96 | 0.164 | 0.165 | 0.165 | 0.165 | 0.001 | 0.163 | -0.001 |
| | 192 | 0.210 | 0.209 | 0.209 | 0.209 | 0.000 | 0.208 | -0.001 |
| | 336 | 0.262 | 0.264 | 0.264 | 0.263 | 0.001 | 0.251 | -0.011 |
| | 720 | 0.341 | 0.342 | 0.342 | 0.341 | 0.000 | 0.339 | -0.002 |
| | Avg. | 0.244 | 0.245 | 0.245 | 0.244 | 0.000 | 0.240 | -0.004 |
| Traffic | 96 | 0.462 | 0.461 | 0.459 | 0.477 | 0.018 | 0.462 | 0.003 |
| | 192 | 0.453 | 0.456 | 0.458 | 0.496 | 0.043 | 0.473 | 0.020 |
| | 336 | 0.475 | 0.475 | 0.461 | 0.516 | 0.054 | 0.498 | 0.037 |
| | 720 | 0.502 | 0.502 | 0.505 | 0.538 | 0.036 | 0.506 | 0.004 |
| | avg. | 0.473 | 0.473 | 0.471 | 0.507 | 0.036 | 0.484 | 0.013 |

in Table 3, but it also demonstrates higher prediction accuracy when trained on the same amount of labeled data. This suggests that CoMRes possesses generalization ability.

The table 10 shows the results of training with distant data from the test period. Compared to Table 3, the overall best scores have changed due to intricate temporal variations. This suggests that the relevance of the data is more important than the amount of data used. As the same as result of the recent data, CoMRes consistently achieves the best performance across all label sizes and prediction horizons compared to the baseline.

## A.8    DISCUSSION - MORE RELATED WORKS

**Semi-supervised learning in other domain** Semi-supervised learning have been extensively studied in domains such as natural language processing (NLP) (Lim et al., 2020; Chen et al., 2020; Sawhney et al., 2021; Park & Caragea, 2024) and image classification (Laine & Aila, 2017; Xie et al., 2020; Zhang et al., 2021; Chen et al., 2023). In these fields, acquiring unlabeled data is relatively straightforward due to the availability of large-scale datasets comprising sentences or images from the web. Ensemble-based loss functions, which effectively leverage unlabeled data, have been shown to enhance model performance by capitalizing on this abundance. Moreover, data augmentation offers a computationally efficient and accessible approach in resource-constrained scenarios. Techniques such as augmentation with pseudo-labels have been particularly effective in expanding training sets for classification tasks, further boosting model performance.

However, extending these approaches to time series forecasting introduces unique challenges. Unlike NLP or image classification, time series data often lacks the diversity and abundance of high-quality unlabeled data. Additionally, the temporal and continuous nature of time series data makes

Table 9: Multivariate time series forecasting results for additional baselines. The input length $L = 96$ ($L = 36$ for the ILI dataset). The best results are highlighted in **bold**. CoMRes's performance represents the average values over 5 runs. The results for TimeMixer are taken from Wang et al. (2024), the results for PatchTST are sourced from Nie et al. (2023), and the remaining results are obtained from Chen et al. (2024).

| Method | | CoMRes | | Pathformer | | PatchTST | | TimeMixer | | Nlinear | | Scaleformer | | TiDE | |
| --- | --- | --- | --- | --- | --- | --- | --- | --- | --- | --- | --- | --- | --- | --- | --- |
| Metric | | MSE | MAE | MSE | MAE | MSE | MAE | MSE | MAE | MSE | MAE | MSE | MAE | MSE | MAE |
| ETTh1 | 96 | 0.378 | **0.386** | 0.382 | 0.400 | 0.394 | 0.408 | **0.375** | 0.400 | 0.386 | 0.392 | 0.396 | 0.440 | 0.427 | 0.450 |
| | 192 | 0.436 | **0.415** | 0.440 | 0.427 | 0.446 | 0.438 | **0.429** | 0.421 | 0.440 | 0.430 | 0.434 | 0.460 | 0.472 | 0.486 |
| | 336 | **0.452** | **0.421** | 0.454 | 0.432 | 0.485 | 0.550 | 0.484 | 0.458 | 0.480 | 0.443 | 0.462 | 0.476 | 0.527 | 0.527 |
| | 720 | **0.474** | **0.453** | 0.479 | 0.461 | 0.495 | 0.474 | 0.498 | 0.482 | 0.486 | 0.472 | 0.494 | 0.500 | 0.644 | 0.605 |
| | Avg. | **0.435** | **0.419** | 0.439 | 0.430 | 0.455 | 0.468 | 0.447 | 0.440 | 0.448 | 0.434 | 0.447 | 0.469 | 0.518 | 0.517 |
| ETTh2 | 96 | **0.275** | **0.327** | 0.279 | 0.331 | 0.294 | 0.343 | 0.289 | 0.341 | 0.290 | 0.339 | 0.364 | 0.407 | 0.304 | 0.359 |
| | 192 | **0.346** | **0.373** | 0.349 | 0.380 | 0.378 | 0.394 | 0.372 | 0.392 | 0.379 | 0.395 | 0.466 | 0.458 | 0.394 | 0.422 |
| | 336 | **0.326** | **0.369** | 0.348 | 0.382 | 0.382 | 0.410 | 0.386 | 0.414 | 0.421 | 0.431 | 0.479 | 0.476 | 0.385 | 0.421 |
| | 720 | **0.393** | **0.419** | 0.398 | 0.424 | 0.412 | 0.433 | 0.412 | 0.434 | 0.436 | 0.453 | 0.487 | 0.492 | 0.463 | 0.475 |
| | Avg. | **0.335** | **0.372** | 0.344 | 0.379 | 0.367 | 0.395 | 0.365 | 0.395 | 0.382 | 0.405 | 0.449 | 0.458 | 0.387 | 0.419 |
| ETTm1 | 96 | **0.310** | **0.339** | 0.316 | 0.346 | 0.324 | 0.361 | 0.320 | 0.357 | 0.339 | 0.369 | 0.355 | 0.398 | 0.356 | 0.381 |
| | 192 | 0.362 | **0.365** | 0.366 | 0.370 | 0.362 | 0.383 | **0.361** | 0.381 | 0.379 | 0.386 | 0.428 | 0.455 | 0.391 | 0.399 |
| | 336 | **0.386** | **0.387** | **0.386** | 0.394 | 0.39 | 0.402 | 0.390 | 0.404 | 0.411 | 0.407 | 0.524 | 0.487 | 0.424 | 0.423 |
| | 720 | 0.456 | **0.424** | 0.460 | 0.432 | 0.461 | 0.438 | 0.454 | 0.441 | 0.478 | 0.442 | 0.558 | 0.517 | 0.48 | 0.456 |
| | Avg. | **0.379** | **0.379** | 0.382 | 0.386 | 0.384 | 0.396 | 0.381 | 0.396 | 0.402 | 0.401 | 0.466 | 0.464 | 0.413 | 0.415 |
| ETTm2 | 96 | **0.163** | **0.245** | 0.170 | 0.248 | 0.177 | 0.260 | 0.175 | 0.258 | 0.177 | 0.257 | 0.182 | 0.275 | 0.182 | 0.264 |
| | 192 | **0.229** | **0.290** | 0.238 | 0.295 | 0.248 | 0.306 | 0.237 | 0.299 | 0.241 | 0.297 | 0.251 | 0.318 | 0.256 | 0.323 |
| | 336 | **0.291** | **0.330** | 0.293 | 0.331 | 0.304 | 0.342 | 0.298 | 0.340 | 0.302 | 0.337 | 0.34 | 0.375 | 0.313 | 0.354 |
| | 720 | **0.369** | **0.378** | 0.390 | 0.389 | 0.403 | 0.397 | 0.391 | 0.396 | 0.405 | 0.396 | 0.435 | 0.433 | 0.419 | 0.410 |
| | Avg. | **0.263** | **0.311** | 0.273 | 0.316 | 0.283 | 0.326 | 0.275 | 0.323 | 0.281 | 0.322 | 0.302 | 0.350 | 0.293 | 0.338 |
| Electricity | 96 | 0.152 | 0.239 | **0.145** | **0.236** | 0.18 | 0.264 | 0.153 | 0.247 | 0.185 | 0.266 | 0.182 | 0.297 | 0.194 | 0.277 |
| | 192 | **0.166** | **0.252** | 0.167 | 0.256 | 0.188 | 0.275 | **0.166** | 0.256 | 0.189 | 0.276 | 0.188 | 0.300 | 0.193 | 0.280 |
| | 336 | **0.181** | **0.267** | 0.186 | 0.275 | 0.206 | 0.291 | 0.185 | 0.277 | 0.204 | 0.289 | 0.210 | 0.324 | 0.206 | 0.296 |
| | 720 | **0.210** | **0.292** | 0.231 | 0.309 | 0.247 | 0.328 | 0.225 | 0.310 | 0.245 | 0.319 | 0.232 | 0.339 | 0.242 | 0.328 |
| | Avg. | **0.177** | **0.263** | 0.182 | 0.269 | 0.205 | 0.290 | 0.182 | 0.273 | 0.206 | 0.288 | 0.203 | 0.315 | 0.209 | 0.295 |
| ILI | 96 | 1.907 | 0.830 | **1.587** | **0.758** | 1.724 | 0.843 | - | - | 2.725 | 1.069 | 0.232 | 0.339 | 2.154 | 0.992 |
| | 192 | 1.538 | 0.755 | **1.429** | **0.711** | 1.536 | 0.752 | - | - | 2.530 | 1.032 | 2.745 | 1.075 | 2.436 | 1.042 |
| | 336 | **1.490** | 0.766 | 1.505 | **0.742** | 1.821 | 0.832 | - | - | 2.510 | 1.031 | 2.748 | 1.072 | 2.532 | 1.051 |
| | 720 | **1.693** | 0.811 | 1.731 | **0.799** | 1.923 | 0.842 | - | - | 2.492 | 1.026 | 2.793 | 1.059 | 2.748 | 1.142 |
| | Avg. | 1.657 | 0.791 | **1.563** | **0.753** | 1.751 | 0.817 | - | - | 2.564 | 1.040 | 2.130 | 0.886 | 2.468 | 1.057 |
| Traffic | 96 | 0.468 | **0.270** | 0.479 | 0.283 | 0.492 | 0.324 | **0.462** | 0.285 | 0.645 | 0.388 | 2.678 | 1.071 | 0.568 | 0.352 |
| | 192 | **0.465** | **0.285** | 0.484 | 0.292 | 0.487 | 0.303 | 0.473 | 0.296 | 0.599 | 0.365 | 0.564 | 0.351 | 0.612 | 0.371 |
| | 336 | **0.493** | **0.295** | 0.503 | 0.299 | 0.505 | 0.317 | 0.498 | 0.313 | 0.606 | 0.367 | 0.57 | 0.349 | 0.605 | 0.374 |
| | 720 | 0.535 | 0.324 | 0.537 | 0.322 | 0.542 | 0.337 | **0.506** | **0.313** | 0.645 | 0.388 | 0.576 | 0.349 | 0.647 | 0.410 |
| | Avg. | 0.490 | **0.294** | 0.501 | 0.299 | 0.507 | 0.320 | **0.485** | 0.298 | 0.624 | 0.377 | 1.097 | 0.530 | 0.608 | 0.377 |
| Weather | 96 | **0.151** | **0.19** | 0.156 | 0.192 | 0.177 | 0.218 | 0.163 | 0.209 | 0.168 | 0.208 | 0.288 | 0.365 | 0.202 | 0.261 |
| | 192 | **0.199** | **0.236** | 0.206 | 0.24 | 0.224 | 0.258 | 0.208 | 0.250 | 0.217 | 0.255 | 0.368 | 0.425 | 0.242 | 0.298 |
| | 336 | **0.244** | **0.275** | 0.254 | 0.282 | 0.277 | 0.297 | 0.251 | 0.287 | 0.267 | 0.292 | 0.447 | 0.469 | 0.287 | 0.335 |
| | 720 | **0.334** | **0.331** | 0.340 | 0.336 | 0.35 | 0.345 | 0.339 | 0.341 | 0.351 | 0.346 | 0.640 | 0.574 | 0.351 | 0.386 |
| | Avg. | **0.232** | **0.258** | 0.239 | 0.263 | 0.257 | 0.280 | 0.240 | 0.272 | 0.251 | 0.275 | 0.436 | 0.458 | 0.271 | 0.320 |

it difficult to generate meaningful augmented data. These challenges necessitate tailored strategies to successfully adopt semi-supervised learning and data augmentation in the context of time series forecasting.

**Semi-supervised learning for Time Series Classification** The concepts of consensus promotion with pseudo-labeling have also been explored in the context of semi-supervised learning for time series classification (Jawed et al., 2020; Liu et al., 2023; Shin et al., 2023; Bae et al., 2024; Liu et al., 2024b). For instance, Jawed et al. (2020) leverage features learned from self-supervised tasks on unlabeled data, while Shin et al. (2023) propose context-attached augmentation to generate augmented instances with varying contexts that preserve the target instance. Liu et al. (2024b) introduce the Scale-Teaching paradigm, which captures discriminative patterns in time series while mitigating noisy labels. Bae et al. (2024) explore consistency regularization by artificially downsampling high-sampling-rate time series to generate augmented versions. These studies demonstrate the effectiveness of leveraging augmented data with pseudo labeling in time series classification.

Despite these advances, the application of semi-supervised learning and consensus promotion to time series forecasting remains underexplored. In this work, we argue that forecasting poses unique challenges distinct from classification tasks, such as the prediction of continuous outputs rather than discrete labels. Additional complexities include long-term temporal dependencies, seasonality, and error propagation in autoregressive settings. These factors make direct application of pseudo-labeling

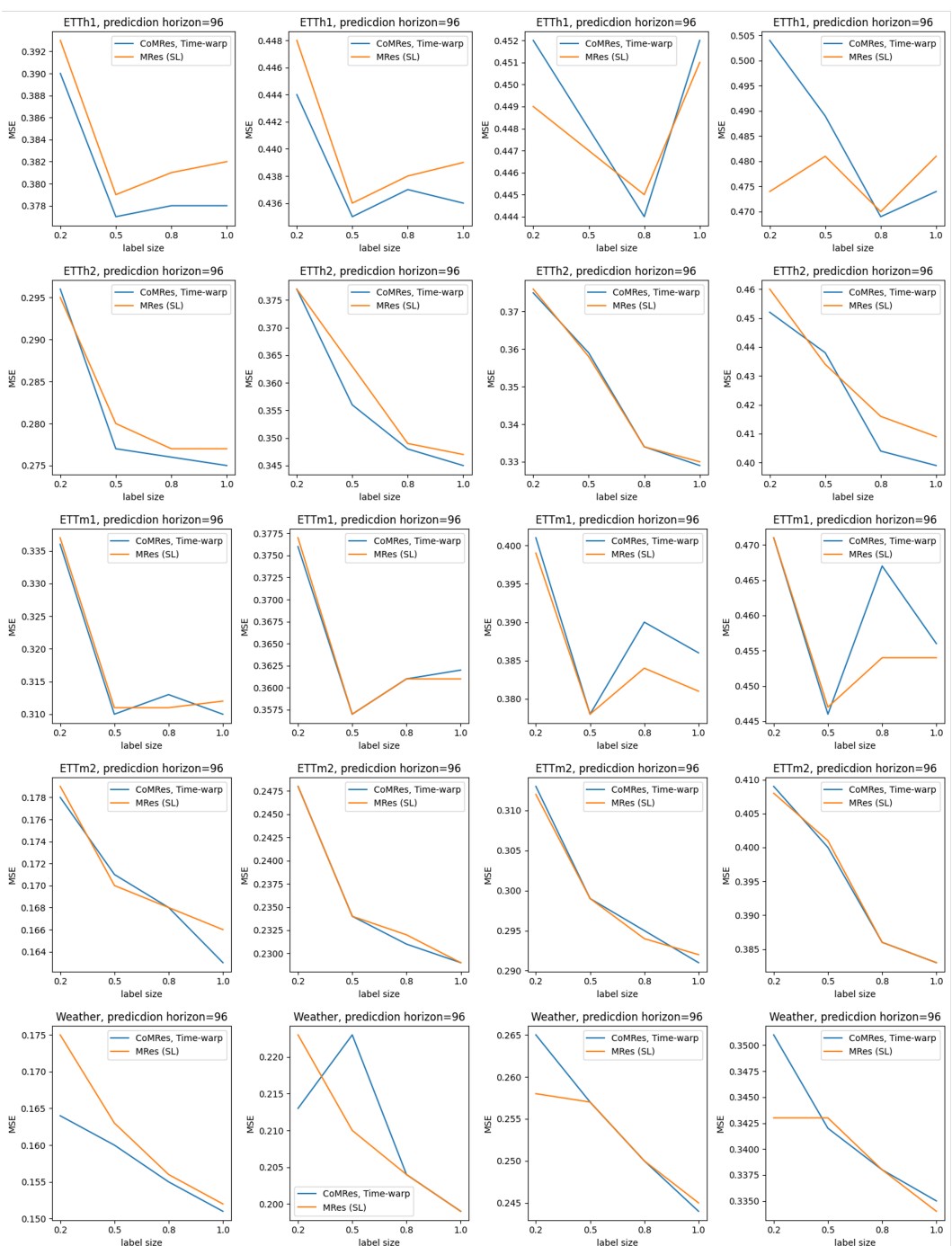

Figure 6: Comparison figures between CoMRes and MRes, evaluated using the same label sizes and dataset

and traditional augmentation techniques problematic, as evidenced by conflicting signals in the lookback context observed during our experiments (Table 1).

To address these challenges, we propose a novel approach that emphasizes consensus promotion on augmented data without relying on pseudo-labels. While our framework may initially seem straightforward, it addresses the non-trivial challenge of integrating ideas from two different fields: semi-supervised learning and time series forecasting. By integrating ideas from semi-supervised learn-

Table 10: Results in Limited-Resource Scenarios: When using the oldest data. The best results among the compared methods are highlighted in bold, while the best result of the remaining methods is underlined. The label size is represented as a ratio of the original training data. For example, a value of 0.5 indicates that only the first half of the original data was used for training, meaning that the most oldest portion of the training data was used.

| Method | | SSL, Time Warp (ours) | | | | SL, Baseline | | | |
|---|---|---|---|---|---|---|---|---|---|
| Label Size | | 0.2 | 0.5 | 0.8 | 1.0 | 0.2 | 0.5 | 0.8 | 1.0 |
| ETTh1 | 96 | 0.412 | 0.398 | **0.378** | **0.378** | 0.409 | 0.396 | 0.381 | 0.382 |
| | 192 | 0.486 | 0.455 | 0.437 | **0.436** | 0.482 | 0.456 | 0.438 | 0.439 |
| | 336 | 0.533 | 0.478 | 0.454 | 0.452 | 0.523 | 0.479 | 0.452 | **0.451** |
| | 720 | 0.579 | 0.502 | 0.506 | **0.474** | 0.544 | 0.506 | 0.484 | 0.481 |
| ETTh2 | 96 | 0.286 | 0.278 | **0.270** | 0.275 | 0.286 | 0.279 | 0.271 | 0.277 |
| | 192 | 0.357 | 0.353 | **0.343** | 0.345 | 0.361 | 0.357 | 0.347 | 0.347 |
| | 336 | 0.343 | 0.334 | 0.333 | **0.329** | 0.346 | 0.338 | 0.333 | 0.330 |
| | 720 | 0.453 | **0.385** | 0.389 | 0.399 | 0.434 | 0.388 | 0.395 | 0.409 |
| ETTm1 | 96 | 0.383 | 0.387 | 0.312 | **0.310** | 0.381 | 0.382 | 0.318 | 0.312 |
| | 192 | 0.412 | 0.416 | 0.368 | 0.362 | 0.412 | 0.416 | 0.371 | **0.361** |
| | 336 | 0.436 | 0.438 | 0.392 | 0.386 | 0.436 | 0.436 | 0.397 | **0.381** |
| | 720 | 0.513 | 0.511 | 0.458 | 0.456 | 0.514 | 0.508 | 0.466 | **0.454** |
| ETTm2 | 96 | 0.180 | 0.172 | 0.166 | **0.163** | 0.180 | 0.170 | 0.168 | 0.166 |
| | 192 | 0.242 | 0.233 | 0.231 | **0.229** | 0.243 | 0.234 | 0.232 | **0.229** |
| | 336 | 0.306 | 0.297 | 0.292 | **0.291** | 0.305 | 0.294 | 0.292 | 0.292 |
| | 720 | 0.409 | 0.390 | 0.386 | **0.383** | 0.406 | 0.389 | 0.384 | **0.383** |

ing and time series forecasting, our framework effectively balances the complexities of forecasting with the benefits of consensus promotion, offering a meaningful contribution.

## A.9 DISCUSSION - RESULT ON ILI DATASET

The ILI dataset presents unique challenges due to its highly non-stationary patterns, which make accurate prediction particularly difficult. While CoMRes demonstrates strong performance across most datasets, its performance on the ILI dataset is less competitive compared to Pathformer, especially when $h = 24$ and $h = 36$, as shown in Table 1. Notably, we were unable to reproduce Pathformer's reported performance on the ILI dataset; however, crediting the authors, Table 1 includes the performance metrics as reported in the original Pathformer paper.

The ILI dataset is characterized by a rapid long-term uptrend, which poses significant challenges for forecasting models. Recent advanced models, such as those proposed by Shabani et al. (2023) and Nie et al. (2023), exhibit high prediction errors (MSE above 2), while others, such as Wang et al. (2024) and Das et al. (2023), exclude the ILI dataset from their experiments, further reflecting its inherent difficulty.

We hypothesize that the rapid long-term uptrend in the ILI dataset limits the advantages of multi-scale modeling, as long-term dependency patterns may not effectively leverage multi-resolution features. Nonetheless, our experimental results show that CoMRes improves prediction accuracy compared to MRes (SL). This suggests that the unsupervised consistency promotion mechanism in CoMRes has a positive impact on enhancing model generalization, even in challenging scenarios like the ILI dataset.

## A.10 DISCUSSION - COMPARISON ON A PER-AUGMENTATION BASIS

To better understand the impact of different augmentation strategies, we conducted a per-augmentation analysis, comparing the performance of CoMRes with MRes across three augmentation techniques: Time Warping, Interpolation, and Noise Injection in Table 11.

The results indicate that the effectiveness of each augmentation depends on both the characteristics of the data and the context in which the augmentation is applied. Augmentations in a supervised learning scenario (e.g., MRes w. augmentation) tend to degrade performance, likely due to overfitting or mismatched data distributions. In contrast, CoMRes utilizes augmented data within a semi-supervised learning framework, which mitigates overfitting and enhances generalization. This distinction is evidenced by CoMRes consistently achieving better results across all tested scenarios.

In conclusion, the per-augmentation analysis highlights the robustness of CoMRes across various augmentation strategies, while demonstrating the critical role of applying augmentations within a semi-supervised framework to achieve consistent and significant improvements over MRes.

Table 11: Results of per-augmentation analysis.

| Method | | Time Warping | | | Interpolation | | | Noise Injection | | | MRes(SL) |
|---|---|---|---|---|---|---|---|---|---|---|---|---|
| | | CoMRes | MRes | Δ | CoMRes | MRes | Δ | CoMRes | MRes | Δ | |
| Consensus Promotion | | ○ | × | | ○ | × | | ○ | × | | × |
| Leveraging Augmented Data | | unlabel | pseudo-label | | unlabel | pseudo-label | | unlabel | pseudo-label | | × |
| ETTh1 | 96 | 0.378 | 0.406 | 0.028 | 0.379 | 0.414 | 0.035 | 0.379 | 0.416 | 0.037 | 0.382 |
| | 192 | 0.436 | 0.455 | 0.019 | 0.436 | 0.459 | 0.023 | 0.436 | 0.440 | 0.004 | 0.439 |
| | 336 | 0.452 | 0.482 | 0.030 | 0.452 | 0.448 | -0.004 | 0.452 | 0.446 | -0.006 | 0.451 |
| | 720 | 0.474 | 0.475 | 0.001 | 0.478 | 0.473 | -0.005 | 0.478 | 0.447 | -0.031 | 0.481 |
| | Avg. | 0.435 | 0.455 | 0.020 | 0.436 | 0.449 | 0.012 | 0.436 | 0.437 | 0.001 | 0.438 |
| ETTh2 | 96 | 0.275 | 0.278 | 0.003 | 0.275 | 0.280 | 0.005 | 0.275 | 0.280 | 0.005 | 0.277 |
| | 192 | 0.347 | 0.355 | 0.008 | 0.346 | 0.355 | 0.009 | 0.346 | 0.352 | 0.006 | 0.348 |
| | 336 | 0.326 | 0.346 | 0.020 | 0.328 | 0.335 | 0.007 | 0.326 | 0.336 | 0.010 | 0.332 |
| | 720 | 0.393 | 0.404 | 0.011 | 0.398 | 0.402 | 0.004 | 0.397 | 0.402 | 0.005 | 0.406 |
| | Avg. | 0.335 | 0.346 | 0.011 | 0.337 | 0.343 | 0.006 | 0.336 | 0.343 | 0.007 | 0.341 |
| ETTm1 | 96 | 0.310 | 0.326 | 0.016 | 0.311 | 0.418 | 0.107 | 0.311 | 0.418 | 0.107 | 0.312 |
| | 192 | 0.362 | 0.367 | 0.005 | 0.364 | 0.430 | 0.066 | 0.364 | 0.436 | 0.072 | 0.361 |
| | 336 | 0.386 | 0.404 | 0.018 | 0.386 | 0.456 | 0.070 | 0.387 | 0.518 | 0.131 | 0.381 |
| | 720 | 0.456 | 0.467 | 0.011 | 0.456 | 0.504 | 0.048 | 0.456 | 0.532 | 0.076 | 0.454 |
| | Avg. | 0.379 | 0.391 | 0.013 | 0.379 | 0.452 | 0.073 | 0.380 | 0.476 | 0.097 | 0.377 |
| ETTm2 | 96 | 0.163 | 0.168 | 0.005 | 0.163 | 0.175 | 0.012 | 0.163 | 0.176 | 0.013 | 0.167 |
| | 192 | 0.229 | 0.230 | 0.001 | 0.229 | 0.240 | 0.011 | 0.229 | 0.241 | 0.012 | 0.231 |
| | 336 | 0.291 | 0.295 | 0.004 | 0.292 | 0.308 | 0.016 | 0.292 | 0.308 | 0.016 | 0.292 |
| | 720 | 0.369 | 0.385 | 0.016 | 0.369 | 0.405 | 0.036 | 0.369 | 0.401 | 0.032 | 0.369 |
| | Avg. | 0.263 | 0.270 | 0.007 | 0.263 | 0.282 | 0.019 | 0.263 | 0.282 | 0.018 | 0.265 |
| Weather | 96 | 0.151 | 0.155 | 0.004 | 0.151 | 0.159 | 0.008 | 0.151 | 0.163 | 0.012 | 0.152 |
| | 192 | 0.199 | 0.203 | 0.004 | 0.200 | 0.205 | 0.005 | 0.200 | 0.208 | 0.008 | 0.199 |
| | 336 | 0.244 | 0.248 | 0.004 | 0.246 | 0.25 | 0.004 | 0.246 | 0.252 | 0.006 | 0.245 |
| | 720 | 0.335 | 0.334 | -0.001 | 0.335 | 0.334 | -0.001 | 0.334 | 0.338 | 0.004 | 0.334 |
| | Avg. | 0.232 | 0.235 | 0.003 | 0.233 | 0.237 | 0.004 | 0.233 | 0.240 | 0.008 | 0.233 |

## A.11 DISCUSSION - COMPUTATION COST

In this section, we analyze the computational efficiency of CoMRes in comparison to Pathformer, with a focus on both the training and inference stages.

**Training Time** Pathformer's training involves operations such as temporal decomposition using the Discrete Fourier Transform (DFT) and weight generation with two learnable parameters. In contrast, CoMRes utilizes $M + 1$ Multi-Layer Perceptrons (MLPs) and integrates data augmentation as a fundamental component of its design. Although CoMRes introduces additional parameters due to the extra MLPs, the capabilities of modern GPUs effectively mitigate potential performance limitations, ensuring the computational requirements remain practical. Runtime metrics confirm that the increased burden is manageable (see Table 12). Notably, the longer training time observed for CoMRes is primarily due to the overhead associated with on-the-fly data augmentation. This additional cost can be reduced by employing precomputed augmentations or optimizing the pipeline through parallelization.

Table 12: Quantitative comparison of training and inference times for CoMRes and Pathformer. Results are based on a lookback window ($L = 96$), prediction horizon ($h = 96$), $M = 4$ views, and identical configurations (e.g., patch sizes) for both models.

| | #parameters | | Train (seconds per epoch) | | | Inference(seconds for all test sample) | | |
|---|---|---|---|---|---|---|---|---|
| | CoMRes | pathformer | batch size | CoMRes | pathformer | #test samples | CoMRes | pathformer |
| ETTh1 | 379,054 | 231,222 | 512 | 4.17 | 3.56 | 2,785 | 1.89 | 1.67 |
| ETTm2 | 2,752,162 | 2,161,962 | 512 | 18.81 | 12.47 | 11,425 | 5.64 | 4.45 |
| Weather | 627,408 | 332,120 | 256 | 47.57 | 33.40 | 10,444 | 7.43 | 7.35 |
| Traffic | 3,212,905 | 2,622,705 | 8 | 478.51 | 269.08 | 3,413 | 37.83 | 32.23 |

**Inference Time** During inference, Pathformer performs a top-$k$ weighted sum operation followed by an MLP to produce its final predictions. In comparison, CoMRes employs an equal-weight sum, which is computationally simpler, followed by an MLP. The computational overhead for CoMRes during inference is minimal, and the additional complexity introduced by its design does not significantly impact overall performance. CoMRes has the flexibility to use the comprehensive prediction directly

as the final output, bypassing the need for $M$ individual MLP computations for each view. This approach further reduces the computational burden in practical scenarios, making CoMRes efficient for inference.

To provide empirical context, we measured the training and inference times for both models under a standardized experimental setup on an NVIDIA A6000 GPU. The results in Table 12 indicate that while computational differences between CoMRes and Pathformer exist, they are manageable and can be further optimized in practical scenarios.

## A.12 VISUALIZATION

We visualize the prediction results of CoMRes and Pathformer on ETTh1, Weather, Traffic dataset. As shown in Figures 7, 8 and 9, the prediction curve closely aligns with the ground truth, demonstrating the excellent predictive performance of CoMRes. Additionally, we present the prediction results of CoMRes, MRes, and Pathformer in an autoregressive forecasting scenario. Figure 10 illustrates that CoMRes effectively mitigates error accumulation.

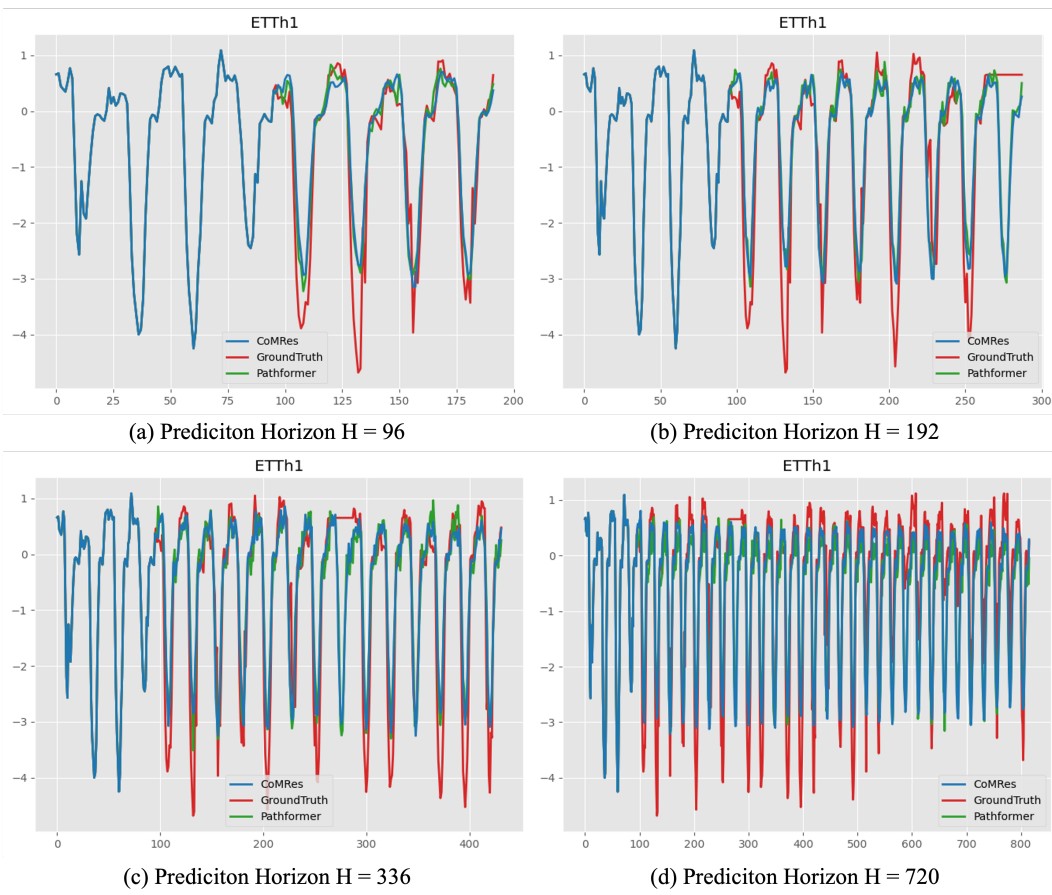

(a) Prediciton Horizon H = 96

(b) Prediciton Horizon H = 192

(c) Prediciton Horizon H = 336

(d) Prediciton Horizon H = 720

Figure 7: Visualization of CoMRes's and Pathformer's prediction results on ETTh1. The input length H = 96

## A.13 LIMITATIONS AND FUTURE WORK

CoMRes has demonstrated favorable performance in long-term time series forecasting; however, there is room for further exploration and refinement of its design options. First, our framework assumes a Euclidean distance to measure consensus between the aggregated prediction and each view. While this approach is straightforward and widely used, it may not be optimal for all scenarios. Alternative metrics, such as soft-DTW (Dynamic Time Warping, Cuturi & Blondel (2017)), which better capture

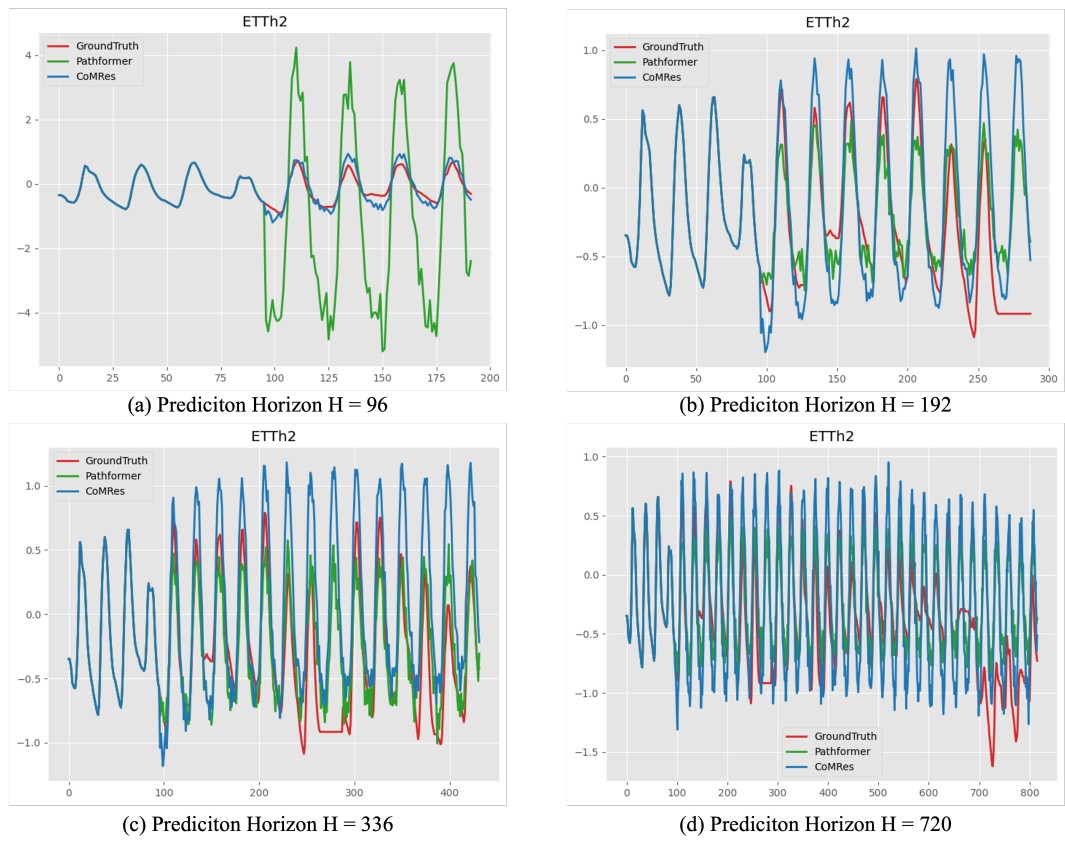

(a) Prediciton Horizon H = 96

(b) Prediciton Horizon H = 192

(c) Prediciton Horizon H = 336

(d) Prediciton Horizon H = 720

Figure 8: Visualization of CoMRes's and Pathformer's prediction results on ETTh2. The input length H = 96

temporal alignment and dependencies, or differentiable loss functions, could be investigated to improve performance. Additionally, this study focuses on three basic augmentation strategies—time warping, noise injection, and interpolation. Although these techniques are effective and simple to implement, they may not fully leverage the potential of more complex augmentation methods. As demonstrated in Section 4.3.2, the effectiveness of augmentation techniques varies depending on the dataset's characteristics. Future work could explore the performance of our framework with advanced augmentation strategies, such as learned augmentations or domain-specific transformations, to provide deeper insights and enhance the applicability and robustness of the proposed approach. Lastly, we employ multi-scale patch division with various patch sizes, which is similar to the model architecture used in Pathformer (Chen et al., 2024). In contrast, other multi-scale models, such as TimeMixer(Wang et al., 2024) and Scaleformer(Shabani et al., 2023), generate multi-scale time series through pooling. Incorporating consensus mechanisms between these pooled series could be a promising direction for extending our framework.

For future research, we aim to explore further improvements in non-Markovian frameworks, where the model would benefit from considering the entire history rather than relying on limited past observations. Another promising direction is investigating how to ensure consistency in multi-scaling predictions across different time horizons, enhancing the overall reliability of forecasts. These studies could further strengthen the capability of our multi-view learning strategy as a tool for long-term time series forecasting across diverse applications.

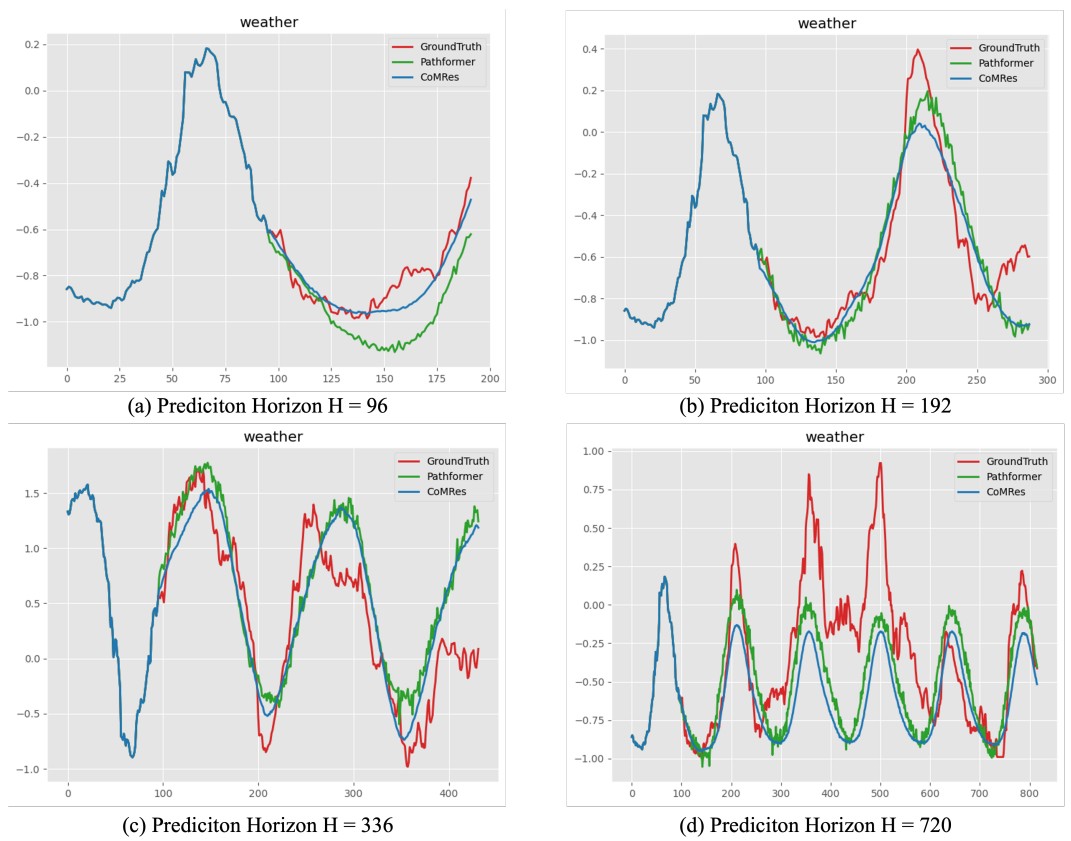

Figure 9: Visualization of CoMRes's and Pathformer's prediction results on Weather. The input length H = 96

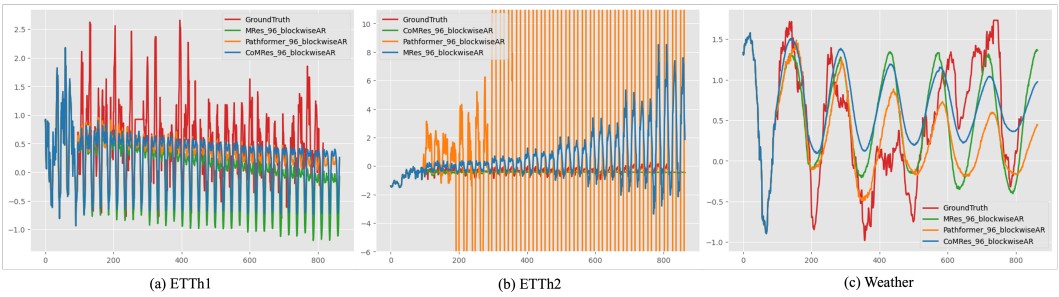

Figure 10: Visualization of CoMRes's, MRes's and Pathformer's autoregressive prediction results on ETTh1, ETTh2 and Weather.

