# OpenReview forum: "CoMRes: Semi-Supervised Time Series Forecasting Utilizing Consensus Promotion of Multi-Resolution"
_ICLR.cc/2025/Conference — ICLR 2025 Poster_

### Official Review · Reviewer_GunG · 2024-10-21

**Soundness:** 3
**Presentation:** 3
**Contribution:** 2
**Rating:** 5
**Confidence:** 4

**Summary:**

This paper discusses multi-resolution long-term time-series forecasting (LTSF). This work extends the Pathformer architecture with two components: data augmentation, and consensus-based training loss for multiple MLP outputs. Comparison between the ablated components and Pathformer is held on widely used time-series prediction benchmarks. Proposed CoMRes improves over the baselines.  Further analysis is done on resource-efficient case.

**Strengths:**

1) The multi-resolution time-series forecasting task is an important direction.

2) Authors have done an in-depth study on the ablation study.

**Weaknesses:**

1) I am uncertain about the novelty of the proposed paper. This paper borrows most of the architecture from the Pathformer, except the MLP for the individual component output and the aggregation layer. Furthermore, the consensus-based loss that minimizes the relative difference can be naturally formulated while utilizing the ensemble. I feel like this paper is a sheer extension of the Pathformer and the previous methods that utilize data augmentation.

2) The experiment results do not convince the efficacy of the method. MRes with consensus training often underperform over the MRes baseline. Furthermore, performance differences between the data augmentations are small. This questions the efficacy of each component as a general method. Finally, ComRes drastically underperforms Pathformer in one dataset, which questions the efficacy of the whole algorithm in general.

3) The extra MLP layer may induce an extra computation burden on the training and inference stage, especially since (M+1) calculations are all required for the final computation. Maybe further analysis of the computation efficiency will further help to understand the limitations.

**Questions:**

See weaknesses.

---

> ### Author Response · Authors · 2024-11-18
> **Response to Reviewer GunG (Part 1)**
>
> Thank you for taking the time to review our paper and for your thoughtful and valuable feedback. Below, we address each of your comments and questions in detail.
>
> > ### [Q1] “the novelty of the proposed paper”
>
> We appreciate the reviewer's feedback and would like to clarify the novelty of our work. Our approach addresses a different aspect of multi-scale (multi-view) modeling and long-term time series forecasting (LSTF), focusing on improving consistency and generalization when encountering unseen data. Individual-view models often produce divergent predictions due to variability in single-scale representations (as demonstrated in **Appendix A.2: Variance of Individual-View Model**). To address this, we enhance agreement among multiple single-view models by leveraging augmented unseen data.
>
> We would like to highlight the core idea of CoMRes that might be helpful to you in understanding the novelty of our paper.
>
> - **Semi-Supervised Approach for Time Series Forecasting**: CoMRes introduces a semi-supervised framework that leverages a multi-view setting on augmented data without assigning corresponding labels. Addressing the effective utilization of unlabeled data in time series forecasting is a relatively unexplored yet crucial area, and we think that our approach paves the way for further research.
>
> - **Impact of Strategies on Labeled and Unlabeled Data**: Our experiments are designed to evaluate the influence of two core strategies—consensus promotion and data augmentation—on both labeled and unlabeled data. CoMRes, which employs consensus promotion on augmented unlabeled data, consistently outperforms the baselines, including:
>   - Augmentation on labeled data (**MRes w. Augmentation**) : CoMRes show superiority in 7 datasets among 8.
>   - Consensus promotion on labeled data (**MRes w. Consensus**): CoMRes show superiority 6 out of 8 datasets, with one dataset yielding comparable results.
>   - Naive supervised learning (**MRes(SL)**) :CoMRes show superiority in 7 datasets among 8.
>   - one of SOTA and base architecture (**Pathformer**, Chen et al., 2024)  : CoMRes show superiority in all 8 datasets.
>
> - **Novel Use of Data Augmentation for Consensus Promotion**: While we include three simple CoMRes variants in our paper for simplicity, the key idea lies in generating unseen data without assigning corresponding labels, rather than developing novel data augmentation techniques. Unlike traditional applications of data augmentation(on supervised loss), which often degrade model performance when used to expand labeled data, we employ existing augmentation techniques in a novel way. Specifically, we use them exclusively for consensus promotion, which has demonstrated significant performance improvements in most of our experiments.
>   - **MRes w. augmentation  vs.  MRes(SL)** : MRes w. augmentation shows inferiority in 6 out of 8 datasets..
>   - **CoMRes vs. MRes w. augmentation** : CoMRes show superiority in 7 datasets among 8.
>   - Related studies, including Wen et al. (IJCAI, 2021), demonstrate that using data augmentation with labeled data can yield negative results for certain datasets in forecasting. Similarly, Semenoglou et al. (Pattern Recognition, 2023) highlight that the effectiveness of data augmentation depends heavily on the specific technique employed.
>
> ### Ref.
>
> Wen et al. Time Series Data Augmentation for Deep Learning: A Survey (IJCAI, 2021)
>
> Semenoglou et al. Data augmentation for univariate time series forecasting with neural networks (Pattern Recognition, 2023)
>
> To better articulate these contributions, we have revised the **introduction** and **Related works** in the revised paper, with changes highlighted in blue.

---

> ### Author Response · Authors · 2024-11-18
> **Response to Reviewer GunG (Part 2)**
>
> > ### [Q2] “The experiment results do not convince the efficacy of the method. MRes with consensus training often underperform over the MRes baseline.”
>
> We believe there may be some misunderstanding regarding the experimental setup. As shown in Table 1, the first three columns (CoMRes) represent our proposed method, while the remaining columns—MRes w. augmentation, MRes w. consensus, MRes (SL)— correspond to baseline methods. These methods are not an existing approach but rather a version derived from CoMRes, created specifically for the ablation study by isolating certain components of the framework. The blue-highlighted delta values indicate that CoMRes consistently outperforms the baselines.
>
> - **"MRes w. consensus" as a Baseline**:  Although it is a version we created, "MRes w. consensus" serves as one of the baseline methods and is not part of the proposed CoMRes framework. This baseline applies consensus promotion exclusively to labeled data. When compared to our model, CoMRes consistently demonstrates better performance across datasets. (CoMRes (ours) vs. MRes w. consensus shows superiority in 6 out of 8 datasets, with one dataset yielding comparable results.)
>
> As noted, consensus promotion is expected to reduce the variance of individual-view models. However, it is not always advantageous for the prediction task. When applied to labeled data, consensus promotion introduces a risk of overfitting, which we believe explains the inconsistent improvement patterns observed with "MRes w. consensus." In contrast, CoMRes applies consensus promotion on augmented data without assigning corresponding labels. This approach enhances the model’s generalization ability and is evidenced by the consistently lower MSE scores achieved by CoMRes across all evaluated datasets.
>
> To enhance clarity, we have elaborated on the performance comparisons between CoMRes and the baseline methods in Section **4.1 Results** and **Table 1** of the revised paper, with the changes clearly highlighted in blue.
>
> > ### [Q2] “ComRes drastically underperforms Pathformer in one dataset, which questions the efficacy of the whole algorithm in general.”
>
> **CoMRes underperforms Pathformer in ILI dataset (h=24, 36)**: We have recognized that CoMRes shows lower performance compared to Pathformer on the ILI dataset. However, it is important to note that we were unable to reproduce Pathformer’s reported performance on this dataset. To ensure fairness and transparency, the performance of Pathformer on ILI dataset is sourced directly from its original paper. The ILI dataset presents unique challenges due to its pronounced non-stationary patterns, which make accurate prediction particularly difficult. We hypothesize that these characteristics contribute to the lower prediction accuracy observed. This hypothesis, along with a detailed discussion of the results on the ILI dataset, is provided in **Appendix 7: Discussion – Results on the ILI Dataset**.
>
> > ### [Q3] The extra MLP layer may induce an extra computation burden.
>
> Our approach focuses on **diversifying the training data** rather than introducing a novel model architecture. It is important to note that other multi-resolution methods, such as Pathformer, TimeMixer, and Scaleformer, rely heavily on their specific architectures, with parameters that are highly sensitive to hyperparameter choices (e.g., the number of hierarchical layers, patch sizes, or pooling scales). As a result, computational burdens cannot be directly compared across methods.
>
> While we produce \( M+1 \) predictions, necessitating an extra MLP layer, we also consider promoting consensus at the **representation level** (before the MLP layer). We think that promoting prediction-level consensus not only yielded marginally superior accuracy compared to representation-level consensus but also provided a more interpretable framework for aligning predictions.
>
> Thank you again for your feedback! We hope our explanations adequately address your questions and concerns. If not, please do not hesitate to let us know—we would be happy to discuss further.

---

> ### Author Response · Authors · 2024-11-23
> **Request of Reviewer's attention and feedback**
>
> Dear Reviewer,
>
> Thanks for your valuable and constructive review, which has inspired us to improve our paper further substantially. This is a kind reminder that it has been 3 days since we posted our rebuttal. Please let us know if our response has addressed your concerns.
>
> Following your suggestions, we have provided the following revisions to our paper:
>
> 1.  Main text - All revisions have been incorporated in the revised paper, with changes highlighted in blue.
> - introduction and related works : Revised to better articulate our contributions, with changes highlighted in blue.
> - 4.1 Time Series Forecasting: To highlight the primary objective of our experiments
>   - Rephrased the "Baseline" section for clarity.
>   - Add the "Ablations" section for clarity.
>   - Elaborated on the performance comparisons between CoMRes and the baseline methods in "Results" section and Table 1.
> 2. Appendix
> - A.2 Variance of Individual-view Model
> - A.7: Discussion – Results on the ILI Dataset
>
> Sincere thanks for your dedication! We are looking forward to your reply.

---

> ### Comment · Reviewer_GunG · 2024-11-25
> **Response to the rebuttal.**
>
> Thank you for your rebuttal.
> I acknowledge that I have read all the reviews and the corresponding responses.
>
> I appreciate the author's response in Q2.
>
> However, my main concerns about the novelty and the computation efficiency have not been resolved.
>
> Unlike the author's proposition, I still think the paper borrows much from Pathformer and only adds meaningful components to the "ensemble-based" loss. Furthermore, the ensemble-based prediction loss has been exhaustively explored [1]. I feel the paper's strategy does not differ from the previous idea except for applying it to a time series.
>
> I stick to my current rating.
>
>
> [1] TEMPORAL ENSEMBLING FOR SEMI-SUPERVISED LEARNING [ICLR 2017]

---

> > ### Author Response · Authors · 2024-11-28
> > **Kindly Seeking Your Feedback on Revised Submission**
> >
> > Dear Reviewer GunG
> >
> > As the rebuttal revision upload deadline is fast approaching, we have updated our revision to address the concerns you raised. Your review has been invaluable in helping us improve our paper, and we sincerely hope this revised version addresses your concerns effectively.
> >
> > - A.7 Discussion - More Related Works
> >   - Semi-supervised learning in other domain
> >   - Semi-supervised learning for Time Series Classification
> > - A.10 Discussion - Computation Cost
> >
> > We kindly request your reconsideration of the score based on this updated submission.
> >
> > Thank you again for your insightful feedback and thoughtful review.

---

> > > ### Comment · Reviewer_GunG · 2024-11-28
> > > **Response to the update.**
> > >
> > > I read the updated paper and the response to the other reviewers as well.
> > >
> > > I am still skeptical about the novelty of the paper. However, I appreciate the author's effort on the rebuttal that cleared concerns on the other comments on the paper.
> > >
> > > I decided to update my score.

---

> > > > ### Author Response · Authors · 2024-12-02
> > > > **Thank you for your kind reply**
> > > >
> > > > Thank you for reviewing our response and for updating your score—we sincerely appreciate your thoughtful and constructive feedback, which has been invaluable in helping us improve our submission. While we respect your perspective on the paper's novelty, we kindly wish to emphasize that our work introduces meaningful advancements in multi-scale modeling for time series forecasting, specifically through consensus promotion and the effective utilization of augmented data.
> > > >
> > > > We want to assure you that we have carefully refined and enhanced the manuscript to address your comments comprehensively. If you have any further questions or concerns, please do not hesitate to let us know—we would be more than happy to discuss them further. Thank you again for your time and thoughtful engagement with our work.

---

> ### Author Response · Authors · 2024-11-27
> **Response II to Reviewer GunG (Part 1)**
>
> Thank you for taking the time to review our work and for your thoughtful feedback. We sincerely appreciate your acknowledgment of our response to Q2 and your engagement with the key aspects of our method.
>
> We understand your continued concerns regarding the novelty of our approach and computational efficiency, and we respect your perspective. However, we would like to kindly offer further clarifications to address the points you have raised and provide additional context on how our contributions go beyond the existing works, including Pathformer and ensemble-based prediction loss techniques like Temporal Ensembling [1].
>
> > ### computation efficiency
>
> We would like to clarify the computational impact of CoMRes (and MRes, which share the same architecture) compared to Pathformer, highlighting the differences between the two approaches for both inference and training stages.
>
> **Training time**
>
> While Pathformer includes operations such as temporal decomposition (Discrete Fourier Transform) and weight generation (with two learnable parameters $W_r$ and $W_{noise}$), CoMRes relies on M MLPs and data augmentation as part of its design.
>
> - CoMRes does involve additional parameters compared to Pathformer due to the extra MLPs; however, with modern GPUs, this has not posed significant limitations in practice. Even with M+1 MLP computations required during training, the computational burden remains manageable, as evidenced by our runtime metrics(see below table).
> - The longer training time of CoMRes is primarily attributed to the overhead introduced by generating augmented data on-the-fly. This can be alleviated by precomputing augmentations or parallelizing the augmentation pipeline.
>
> **Inference time**
>
> Pathformer involves a top-k weighted sum followed by an MLP for its comprehensive prediction, whereas CoMRes uses an equal-weight sum followed by an MLP.
>
> - The computational difference is marginal during inference, as the additional overhead for CoMRes is minimal compared to the weighted sum operation in Pathformer.
> - CoMRes has the flexibility to use the comprehensive prediction directly as the final output, bypassing the need for M individual MLP computations for each view. This approach further reduces the computational burden in practical scenarios, making CoMRes efficient for inference.
>
> To provide context, we measured the approximate training and inference times under our standard experimental setup (an NVIDIA A6000):
> |         |   #parameter  |            |  |  Train (seconds per epoch)   |                   |  | Inference(seconds for all test sample)         |            |
> |---------|:-------------:|:----------:|:----------:|:------:|:----------:|:---------------:|:------:|:----------:|
> |         | **CoMRes**        | **pathformer** | **batch size** | **CoMRes** | **pathformer** | **#test samples**     | **CoMRes** | **pathformer** |
> | **ETTh1**   |  379,054        | 231,222    | 512        | 4s  | 3s         | 2,785            |  1.8s  |    1.6s    |
> | **ETTm2**   | 2,752,162   |   2,161,962      | 512        | 18s    | 12s        | 11,425           |  5.6s  |    4.4s    |
> | **Weather** |    627,408     |    332,120  | 256        | 47s    | 33s        | 10,444           |  7.4s  |    7.3s    |
> | **Traffic** |     3,212,905     |  2,622,705   | 8        | 478s | 269s        | 3,413           |  37.8s  |    32.2s    |
>
> #### *note*  :  In this setting, lookback L=96, prediction horizon h=96, with M=4 and identical configuration(e.g. patch sizes) for both CoMRes and Pathformer
>
> We believe that the **computational differences between CoMRes and Pathformer, while present, do not present a significant limitation in practical scenarios**.

---

> ### Author Response · Authors · 2024-11-27
> **Response II to Reviewer GunG (Part 2)**
>
> > ### novelty
>
> We greatly appreciate the opportunity to clarify the novelty of our approach and its adaptation to the distinct challenges of time series forecasting. While it is true that ideas similar to consensus promotion in semi-supervised learning and data augmentation have been explored in other "classification" domains, such as natural language processing (NLP) and image classification, notably in the work you referenced [1], time series "forecasting" presents distinct challenges that make the application of these ideas far from straightforward.
>
> **On Comparisons to Temporal Ensembling and Prior Works**
>
> The idea of ensemble-based loss and consensus promotion has been explored in the context of semi-supervised learning for time series classification such as in the works of Liu et al. (2024), Bae et al. (2024) and others [2-6]. These works indeed offer valuable insights into leveraging augmented data in time series classification tasks. However, we argue that applying semi-supervised learning and consensus promotion to time series “forecasting” is an unexplored area and we believe we are the first to show consistent improvement across numerous datasets utilizing semi-supervised learning techniques.
>
> The application of such techniques to time series forecasting—which is inherently a regression task with continuous outputs—poses different challenges to previous problems. In classification, it is generally acceptable to predict the same label for inputs that appear similar. However, in time series forecasting (or regression tasks), assigning identical values to different inputs with similar lookbacks can confuse the model. Our experience confirms this challenge. When we initially experimented with straightforward data augmentation techniques in time series forecasting (e.g., MRes with consensus), the improvements were limited and less consistent than expected. We attribute this to the conflicting signals introduced by augmentations applied to the same context (lookback), which can lead to instability in predictions. This motivated the core contribution of our approach, which leverages unseen data to refine predictions without relying on pseudo-labels, avoiding the potential for conflict in learning signals.
>
> We acknowledge that similar techniques have been explored in other domains. However, the fundamental differences between domains—classification versus forecasting—and the specific challenges associated with time series data, necessitate novel adaptations. For example, in time series forecasting, unique factors such as multi-scale and temporal dependencies, long-term trends, seasonality, non-stationary behaviors, the need to generate future values for augmented data, and addressing uncertainties in predictions over extended horizons introduce complexities that are absent in classification tasks. While our proposed approach may appear straightforward at first glance, it addresses the non-trivial challenge of stitching together ideas from two fundamentally different fields. By adapting semi-supervised consensus promotion to the context of time series forecasting, our framework effectively balances the complexities of forecasting with the strengths of consensus promotion on augmented data, offering a significant contribution to the field.
>
> **Ref.**
> [1] Laine, Samuli, and Timo Aila "Temporal Ensembling for Semi-Supervised Learning" (ICLR 2017)
>
> [2] Liu, Zhen, et al. "Scale-teaching: robust multi-scale training for time series classification with noisy labels." (Neurips 2024)
>
> [3] Bae, Minyoung, et al. "Semi-supervised learning for time series collected at a low sampling rate." (SIGKDD 2024)
>
> [4] Liu, Zhen, et al. "Temporal-frequency co-training for time series semi-supervised learning." (AAAI, 2023)
>
> [5] Shin, Yooju, et al. "Context consistency regularization for label sparsity in time series." (ICML 2023).
>
> [6] Jawed, Shayan, et al. "Self-supervised learning for semi-supervised time series classification." (PAKDD 2020, Springer International Publishing, 2020).
>
> **On Comparisons to Pathformer**
>
> It is important to emphasize that this is not primarily an architecture paper. Our focus was not on proposing a novel architecture but on stripping unnecessary components from Pathformer, such as temporal decomposition and weight sampling during aggregation, which did not significantly improve performance in our experiments. It allowed us to simplify the model and focus on the core contributions of our method.
> The architectural differences between MRes (SL) and Pathformer (e.g., replacing adaptive-weighted MLPs with averaging MLPs) account for some performance improvements. However,  the additional performance gains in CoMRes are driven by consensus promotion on the augmented data.

---

> ### Author Response · Authors · 2024-11-27
> **Response II to Reviewer GunG (Part 3)**
>
> **Tackling Error Propagation in Autoregressive Forecasting**
>
> CoMRes also addresses the critical issue of error propagation in autoregressive forecasting, a challenge unique to forecasting tasks. While most time forecasting approaches are limited to predicting a fixed horizon in a single step, we address a more realistic use case: autoregressive forecasting, where the horizon of interest can vary dynamically. In practical scenarios, training models for every possible horizon is neither feasible nor efficient. To address this, one can consider using models trained in autoregressive inference settings. However, in autoregressive settings, prediction errors inevitably accumulate over time, making it increasingly challenging to maintain consistent and accurate predictions. This issue highlights the need for models that can effectively handle error propagation over extended horizons, a key focus of CoMRes. Unlike existing long-term forecasting models, including Pathformer, which primarily focus on predictions within a fixed trained horizon, we go further and examine the model in autoregressive settings. Our analysis of error propagation (Figure 3 and Section 4.2) demonstrates the ability of CoMRes to mitigate this challenge effectively. We believe this capability not only improves forecasting accuracy over longer horizons but also delivers practical value beyond what models like Pathformer can offer.
>
> We hope this clarification highlights the distinct contributions of our work and its practical significance in advancing time series forecasting methods. Thank you once again for your time and feedback—it has been invaluable in refining and strengthening our work.

---

### Official Review · Reviewer_iocg · 2024-11-01

**Soundness:** 3
**Presentation:** 3
**Contribution:** 3
**Rating:** 6
**Confidence:** 4

**Summary:**

This paper proposes a "consensus promotion" learning objective to enhance the consistency of multi-scale time series data predictions. Additionally, given this learning objective is well-suited for semi-supervised learning, the authors introduce data augmentation strategies aligned with the proposed framework to further improve model generalizability. Extensive experiments are conducted on various datasets and ablation settings.

**Strengths:**

(1) This paper proposes and investigates a novel perspective on improving time-series prediction tasks: the consistency of predictions across different patch sizes of time-series data, which represents an interesting and promising research direction.

**Weaknesses:**

The main weakness of this paper lies in its evaluation.

(1) The variance of model performance and metrics is not reported for any experimental results, making it difficult for readers to assess whether the minor improvements are consistent and significant or simply due to stochastic gradient descent. The authors are encouraged to report variances for all results and conduct comprehensive statistical tests to demonstrate whether the improvements are statistically significant.

(2) There is no experimental evidence to demonstrate whether the proposed consensus promotion is actually beneficial. For example, what is the exact prediction MSE for each "individual-view model"? Do they actually become more consistent after applying the unsupervised consistency loss? Additionally, in Table 1, if I understand correctly, the only difference between "MRes (SL)" and "MRes w. consensus" is that "MRes w. consensus" includes additional consensus promotion learning objectives. However, there are no consistent patterns indicating which model's MSE is generally smaller. The authors are encouraged to design clear experiments that demonstrate whether the proposed consensus promotion works as expected and whether it is beneficial for the prediction task.

**Questions:**

N/A

---

> ### Author Response · Authors · 2024-11-18
> **Response to Reviewer iocg**
>
> Thank you for taking the time and providing thoughtful and valuable feedback. Below, we address each of your comments and questions in detail.
>
> > ### [W1] “The authors are encouraged to report variances for all results”
>
> Thanks for the reviewer's valuable suggestion. We have provided the standard deviation results in **Appendix A.4: Error bars**.
>
> > ### [W2]  “what is the exact prediction MSE for each "individual-view model"? Do they actually become more consistent after applying the unsupervised consistency loss?”
>
> In **Table 1**, we present the ensemble prediction MSE for CoMRes and MRes, where CoMRes demonstrates superior performance. We believe that comparing the exact prediction MSE for each individual-view model is not strictly necessary as individual-view models are designed to learn representations in the context of specific temporal resolutions. Instead, we have analyzed the **Variance of Individual-View Models**, as provided in **Appendix A.2**.
>
> **Figure 5 in Appendix A.2** clearly shows that applying the unsupervised consistency loss in CoMRes improves consistency between individual-view models compared to MRes (SL), offering evidence of the effectiveness of our approach.
>
> > ### [W2] “if I understand correctly, the only difference between"MRes (SL)" and "MRes w. consensus" is that "MRes w. consensus" includes additional consensus promotion learning objectives. However, there are no consistent patterns indicating which model's MSE is generally smaller.”
>
> We believe there may be some misunderstanding regarding the experimental setup. As shown in Table 1, the first three columns (CoMRes) represent our proposed method, while the remaining columns—MRes w. augmentation, MRes w. consensus, MRes (SL)— correspond to baseline methods. These methods are not an existing approach but rather a version derived from CoMRes, created specifically for the ablation study by isolating certain components of the framework. The blue-highlighted delta values indicate that CoMRes consistently outperforms the baselines.
>
> **"MRes w. consensus" as a Baseline**:  Although it is a version we created, "MRes w. consensus" serves as one of the baseline methods and is not part of the proposed CoMRes framework. This baseline applies consensus promotion exclusively to labeled data. When compared to our model, CoMRes consistently demonstrates better performance across datasets. (CoMRes (ours) vs. MRes w. consensus shows superiority in 6 out of 8 datasets, with one dataset yielding comparable results.)
>
> As noted, consensus promotion is expected to reduce the variance of individual-view models. However, it is not always advantageous for the prediction task. When applied to labeled data, consensus promotion introduces a risk of overfitting, which we believe explains the inconsistent improvement patterns observed with "MRes w. consensus."
> In contrast, CoMRes applies consensus promotion on augmented data without assigning corresponding labels. This approach enhances the model’s generalization ability and is evidenced by the consistently lower MSE scores achieved by CoMRes across all evaluated datasets.
>
> > ### [W2] “The authors are encouraged to design clear experiments that demonstrate whether the proposed consensus promotion works as expected and whether it is beneficial for the prediction task.”
>
> Our experiments are designed to evaluate the influence of two core strategies—consensus promotion and data augmentation—on both labeled and unlabeled data. CoMRes, which employs consensus promotion on augmented unlabeled data, consistently outperforms the baselines, including:
>
> - Augmentation on labeled data (**MRes w. Augmentation**) : CoMRes show superiority in 7 datasets among 8.
> - Consensus promotion on labeled data (**MRes w. Consensus**): CoMRes show superiority 6 out of 8 datasets, with one dataset yielding comparable results.
> - Naive supervised learning (**MRes(SL)**) :CoMRes show superiority in 7 datasets among 8.
> - one of SOTA and similar architecture (**Pathformer**, Chen et al., 2024)  : CoMRes show superiority in all 8 datasets.
>
> _(Superiority is based on the average MSE of each dataset, calculated as the average across 4 different prediction lengths)._
>
> Following the reviewer’s suggestions, we have:
> 1) Rephrased the **Baselines section in Main Text 4.1** for clarity.
> 2) Elaborated on the performance comparisons between CoMRes and the baseline methods in Section **4.1 Results and Table1**.
>
> All revisions have been included in the revised paper, with changes highlighted in blue.
>
> Thank you again for your feedback! We hope our explanations adequately address your questions and concerns. If not, please do not hesitate to let us know—we would be happy to discuss further.

---

> ### Author Response · Authors · 2024-11-23
> **Request of Reviewer's attention and feedback**
>
> Dear Reviewer,
>
> Thanks for your valuable and constructive review, which has inspired us to improve our paper further substantially. This is a kind reminder that it has been 3 days since we posted our rebuttal. Please let us know if our response has addressed your concerns.
>
> Following your suggestions, we have provided the following revisions to our paper:
>
> 1. Main text - All revisions have been incorporated in the revised paper, with changes highlighted in blue.
> - introduction and related works : Revised to better articulate our contributions, with changes highlighted in blue.
> - 4.1 Time Series Forecasting: To highlight the primary objective of our experiments
>   - Rephrased the "Baseline" section for clarity.
>   - Add the "Ablations" section for clarity.
>   - Elaborated on the performance comparisons between CoMRes and the baseline methods in "Results" section and Table 1.
>
> 2. Appendix
> - A.2 Variance of Individual-view Model
> - A.4 Error bars
>
> Sincere thanks for your dedication! We are looking forward to your reply.

---

> > ### Comment · Reviewer_iocg · 2024-11-25
> >
> > Dear Authors,
> >
> > Thank you for your detailed response.
> >
> > (1) The std results in Table 7 confirm the concerns I mentioned in [W1]. Some minor differences fall within one std range. Without p-values, it is difficult to evaluate whether these minor differences are statistically significant, which is important for determining which models are "better" or "no difference".
> >
> > (2) I respectfully disagree with the statement that "comparing the exact prediction MSE for each individual-view model is not strictly necessary." The equation for $L_u$ on line 199 explicitly encourages each individual-view model to make their predictions close to the comprehensive prediction. I believe it is fundamental to clearly demonstrate whether this loss function works as intended through experiments by examining the changes in each individual-view-model's predictions. Besides, in Fig. 5, why is the comparison focused on the variance of CoMRes? CoMRes includes both the consensus promotion and data augmentation modules, right? Why not focus instead on the variance comparison between MRes (SL) and MRes with consensus, where $L_u$ should be the only difference?
> >
> > (3) I appreciate the discussion on the trade-off of consensus promotion between reducing the variance of individual-view models and the risk of overfitting. Should we also consider the risk of overfitting when applying it with data augmentation? What would be the rule of thumb for determining $w_u$?

---

> ### Author Response · Authors · 2024-11-26
> **Response II to Reviewer iocg (Part 1)**
>
> Thank you for your attention and for providing valuable feedback again.
>
> > ### (1) p-values analysis
>
> Thank you for highlighting the importance of statistical significance in evaluating model performance. To address this, we have  **revised the manuscript to include a p-value analysis**, where statistically significant differences at the 95% confidence level are **marked with an asterisk (*) in Table 7**. This addition will provide a clearer indication of which differences are statistically meaningful.
>
> We acknowledge that some minor differences, which are statistically not significant, do exist. Nevertheless, we emphasize that CoMRes consistently outperforms the baseline and the ablation models across datasets and augmentation strategies. This **consistent trend** underscores the practical effectiveness of our approach, even before formal statistical testing.
>
> **Robustness of CoMRes:**
>
> We also believe that robustness is a critical aspect to consider when evaluating models. The robustness of CoMRes compared to the baselines and MRes with ablation components is demonstrated through two key aspects:
>
> - **Size of the Standard Deviation**:
>   -The size of the standard deviation is an important measure of robustness. Compared to Pathformer, CoMRes demonstrates **smaller standard deviations**, indicating greater stability and reliability across datasets and configurations.
>
>
> - **Reduced Error Propagation**:
>   - We analyzed **error propagation over forecasting steps** as an additional evaluation aspect. As shown in **Figure 3**, while prediction error naturally increases along the horizon due to the intrinsic dynamics of time series, CoMRes consistently outperforms baseline models, especially over longer horizons. The results in **Section 4.2: Error Accumulation in Autoregressive Forecasting** further highlight CoMRes’s superior predictive accuracy and reduced error accumulation.
>
>
> These factors collectively demonstrate the **robustness and effectiveness** of CoMRes in addressing the challenges of long-term time series forecasting. We hope these clarifications address your concerns and provide a more comprehensive perspective on the strengths of our method.
>
> > ### (2) whether $L_u$ loss function works as intended
>
> Thank you for your thoughtful feedback and for raising these important points.
>
> **On  examining the changes in each individual-view-model's predictions**
>
> We acknowledge that comparing MRes (SL) with MRes w. consensus would isolate the effect of L_u​ more directly. To address this, we have updated variance of Individual-view analyses in the revised manuscript (**Figure 5 in Appendix A.2**) The Figure 5 provides clear evidence that **applying consistency loss improves consistency between individual-view models** compared to MRes (SL). MRes with consensus and CoMRes exhibits a consistently smoother and lower trend than MRes(SL), indicating reduced variability in predictions.
>
> **On examining exact prediction MSE for each individual-view model's predictions**
>
> We appreciate your suggestion about conducting an ablation study to examine whether the loss function $L_u$ works as intended for individual-view models, and we fully agree with the importance of clarifying this aspect. However, it is essential to note that our goal was not to minimize the MSE for each individual view (i.e., comparing individual predictions directly to test data), but rather to reduce the variability between the views.
>
> In our framework, we aim for multiple views to coordinate effectively and produce consistent predictions, especially when encountering new, unseen data. A high variance among individual views would undermine this objective. By leveraging consensus promotion without requiring explicit future values as labels, our intention was to reduce this variance across individual views in facing unseen data, ensuring that the multi-view models align more closely.
>
> To this end, we believe that the variance of individual-view models serves as a robust proxy for alignment. A reduction in variance between individual-view models (on unseen validation and test sets) directly reflects the impact of $L_u$, as it measures how consistently the models converge toward a consensus. This is why we chose to focus on variance, as it is more aligned with the primary objective of consensus promotion—**reducing divergence between views rather than independently improving the individual-view MSE**.

---

> ### Author Response · Authors · 2024-11-26
> **Response II to Reviewer iocg (Part 2)**
>
> (continue) It is also worth noting that since CoMRes applies consensus promotion on unlabeled data, it does not explicitly aim to reduce the error between individual predictions and the ground truth. Similarly, MRes with consensus applies both supervised loss and consistency loss on labeled data, but the balance between these losses means that individual prediction errors may not always decrease.
>
> Therefore, to address your concerns, we submit the multi-view variance (Appendix A.2) as a part of the ablation study to further clarify its purpose in supporting the objectives of consensus promotion. We hope this provides greater clarity on how our framework is designed to function.
> > ### CoMRes includes both the consensus promotion and data augmentation modules
>
> In CoMRes, consensus promotion is specifically applied to augmented data, which is a key component of our semi-supervised framework. Augmented data is primarily used for the lookback sequence (e.g. $x_{t-L}, \\ldots, x_{t-1}$), while we do not rely on the pseudo-gold prediction($x_t$) from the augmented data.
>
> In a supervised setting, the model typically learns to predict the gold label effectively after a few training epochs. As a result, applying consensus promotion directly to supervised data may offer limited additional benefit and could even lead to diminishing returns. By contrast, applying consensus promotion to augmented data leverages unseen patterns and diverse perspectives, which enhances the model's generalization ability.
>
> We believe this approach is important because it allows the model to effectively learn from both labeled and unlabeled data while mitigating risks such as overfitting or mismatched distributions in training. This design also ensures that consensus promotion contributes meaningfully to aligning predictions across views, even for novel or unseen data scenarios..
>
> > ### (3) the risk of overfitting when applying data augmentation
>
> Thank you for bringing up this important point regarding the potential risk of overfitting when applying data augmentation. While data augmentation is a widely used technique in deep learning, we address that its application in time series forecasting is not as straightforward. Augmented data is artificial and unseen, and using shifted augmented values from the time window as pseudo-labels can introduce noise, potentially degrading performance.
>
> This challenge is demonstrated in our experimental results (Table 1), where MRes w. augmentation (which uses augmented data with pseudo-labels) often leads to a decline in prediction accuracy:
>
> - To address concerns about varying augmentation strategies, we newly included a **per-augmentation analysis (Appendix A.8)** that evaluates performance across different strategies, including Time Warping, Noise Injection and Interpolation. The results consistently show that leveraging augmented data in a semi-supervised setting enhances prediction accuracy and generalization, even when applied across diverse augmentation methods.
>
> Moreover, related studies support these findings:
>
> - Wen et al. (IJCAI, 2021) highlight that using data augmentation with labeled data can yield negative results for certain datasets in forecasting.
> - Semenoglou et al. (Pattern Recognition, 2023) emphasize that the effectiveness of data augmentation depends heavily on the specific technique employed, further demonstrating the need for careful consideration in time series forecasting.
>
> We hope these results and references clarify the risks of overfitting when applying data augmentation in time series forecasting and underscore the robustness of our semi-supervised approach in addressing these challenges.
>
> #### **Ref.**
> Wen et al. Time Series Data Augmentation for Deep Learning: A Survey (IJCAI, 2021)
>
> Semenoglou et al. Data augmentation for univariate time series forecasting with neural networks (Pattern Recognition, 2023)
>
> **On the rule of thumb for determining $w_u$**
>
> We use a warm-up strategy for $w_u$, the weight for the unsupervised consistency loss. In our implementation, $w_u$ is initialized to 0 and gradually increases as training progresses. Specifically, $w_u$ starts increasing from a specified epoch (set to 20 in our experiments) and scales linearly.
> This gradual scaling ensures that the model initially focuses on optimizing the supervised loss before progressively incorporating the unsupervised consistency loss. By doing so, the model effectively balances the supervised and unsupervised objectives. This approach allows the model to first capture essential patterns from labeled data before leveraging the augmented data, promoting better generalization and stability during training.
>
>
> Thank you again for your feedback! We hope our explanations adequately address your questions and concerns.

---

> > ### Comment · Reviewer_iocg · 2024-11-26
> >
> > I appreciate the authors' efforts in precisely evaluating their methods in the revisions. Many of my concerns have been well-addressed, and I have increased my score accordingly.

---

> > > ### Author Response · Authors · 2024-11-28
> > > **Thank you for your kind reply**
> > >
> > > We sincerely thank the reviewer for the positive feedback on our responses and for increasing the score. The revised paper, including the additional analysis you pointed out, has been uploaded. We deeply appreciate your valuable comments, which have significantly enhanced the quality of the manuscript.
> > >
> > > Please feel free to reach out if you have any further questions or concerns.
> > >
> > > Thank you once again for your time and effort.

---

### Official Review · Reviewer_GeBU · 2024-11-02

**Soundness:** 3
**Presentation:** 2
**Contribution:** 2
**Rating:** 6
**Confidence:** 5

**Summary:**

The authors propose CoMRes, a long-term forecasting architecture based on Pathformer, which leverages a consensus promotion mechanism for multi-resolution views. The core idea is to align the supervised predictions of multiple resolutions with the aggregated prediction. Furthermore, they use augmentation strategies (Time Warping, Interpolation and Noise Injection) to construct the different views.  The work also presents an evaluation using autoregressive and non-autoregressive decoding schemes, aiming to show the robustness of the forecasts in longer prediction horizons. The authors claim that CoMRes outperforms traditional supervised models, exhibiting greater robustness and accuracy on multiple popular datasets.

**Strengths:**

- The paper introduces an interesting concept of promoting consistency among multi-resolution views, which has potential in capturing complex temporal relationships.

- The amount of datasets used in the experimental section is exhaustive, which adds impact to the of the experimental results.

- The addition of limited-resource scenarios is valuable, as this is often the case in time series applications.

- In general i think the writing of the paper is good and understandable.

- The discussed related work seems to be exhaustive.

**Weaknesses:**

- The title and abstract do not clearly explain why the proposed approach is semi-supervised. It is challenging to understand how forecasting can be performed in a "semi-supervised" manner, given that "labels" can be constructed by shifting the time window. It would be helpful to provide more concrete explanations on this aspect, particularly regarding the proposed use of augmented data and the challenges with temporal dynamics.

- In lines 031-033, the statement about the superiority of transformers in forecasting is misleading. Multiple recent works (Das et al., 2023; Wang et al., 2024; Zeng et al., 2023) have shown that transformers are not always state-of-the-art. This statement should be reconsidered or at least clarified with respect to specific contexts or benchmarks.

- The comparison against relevant baselines is lacking. The paper does not include benchmarks like TimeMixer, TiDE, PatchTST, D/NLinear, and other advanced transformer-based models (e.g., iTransformer). Including these baselines would significantly strengthen the evaluation and provide a better context for the proposed model.

- The tables, particularly Table 1, are difficult to read. I'm not sure what red or blue means at least its not stated in the caption.

- The baseline method achieves the best result multiple times without the proposed augmentations, which weakens the argument for using the augmentations. In general I'm pretty confused about what the idea of multiple resolutions vs augmentations is in this work.

- While the limited-resource scenario is a useful addition, it does not effectively demonstrate how the method performs on datasets with many variables (e.g., Traffic or Electricity). This should at least be stated in the limitations/conclusion section.

## Refs
Das et al., Long-term forecasting with TiDE: Time-series dense encoder. (TMLR 2023)

Wang et al., TimeMixer: Decomposable multiscale mixing for time series forecasting. (ICLR 2024)

Zeng et al., Are transformers effective for time series forecasting? (AAAI 2023)

**Questions:**

1. In line 032, the paper cites Wang et al., suggesting multi-resolution methods "overlook consistency." Could you provide specific details about what was missed in the multi-resolution reconciliation methods used by Wang et al.?

2. In Section 3, you mention that augmentations are less effective in forecasting compared to classification and anomaly detection. Could you elaborate on why this is the case? Specifically, how does your chosen augmentations differ?

3. Why do you assume a Euclidean distance to measure consensus between the aggregated prediction and each view? Would other measures, such as soft-DTW, work better?

4. Does your framework also work with more "complex" augmentation strategies beyond the three mentioned (time warping, noise injection, interpolation)? Maybe discussing the limitations is a valuable contribution.

5. Could you clarify whether MRes is your method (with ablated components)? I would not count that as a baseline.

6. I'm a bit confused, at what points do you use augmentations, and at what different resolutions (scales)? Could you describe this process in more detail?

7. The paper claims superior robustness compared to traditional supervised learning approaches. Could you clarify how exactly robustness was demonstrated?

---

> ### Author Response · Authors · 2024-11-18
> **Response to Reviewer GeBU (Part 1)**
>
> Thank you for taking the time and providing thoughtful and valuable feedback.
>
> First, we would like to correct your summary:
>
> > ### [Summary]: "The core idea is to align the supervised predictions of multiple resolutions with the aggregated prediction." It  is also related to [W1] "why the proposed approach is semi-supervised".
>
> We would like to highlight the core idea of CoMRes to help clarify our proposed method:
>
> - **Semi-Supervised Approach for Time Series Forecasting**:  CoMRes introduces a semi-supervised framework that leverages a multi-view setting on augmented data without requiring explicit future values as labels. Applying consensus promotion to supervised data alone can lead to overfitting, diminishing the effectiveness of consensus promotion. This is demonstrated by the superior performance of the semi-supervised approach compared to the supervised approach in our experiments.
>
> - **Joint Loss with Unsupervised Consistency Loss**: CoMRes employs consensus promotion, which aligns not only the supervised predictions on labeled data but also the unsupervised predictions on augmented, unseen data.
>
> Thus, with the unsupervised loss, we can train the model on augmented data without requiring corresponding future values. While "labels" for augmented data can theoretically be constructed by shifting the time window, as you mentioned, we believe this process is not straightforward in forecasting. Augmented data is artificial and unseen, and simply using shifted values from the time window as labels introduces potential noise.
>
> Moreover, as demonstrated in our experimental results(Table 1), MRes w. augmentation (which uses augmented data with corresponding labels) often leads to a decline in prediction accuracy.
>   - MRes w. augmentation  vs.  MRes(SL) : MRes w. augmentation shows inferiority in 6 out of 8 datasets..
>   - CoMRes(ours) vs. MRes w. augmentation : CoMRes show superiority in 7 datasets among 8.
>   - Related studies, including Wen et al. (IJCAI, 2021), demonstrate that using data augmentation with labeled data can yield negative results for certain datasets in forecasting. Similarly, Semenoglou et al. (Pattern Recognition, 2023) highlight that the effectiveness of data augmentation depends heavily on the specific technique employed.
>
> This finding supports our belief that using augmented data in supervised learning is less effective and may even hinder performance. Addressing the effective utilization of unlabeled data in time series forecasting is a relatively unexplored but critical area, and we believe our approach paves the way for further research.
>
> ### Ref.
> - Wen et al. Time Series Data Augmentation for Deep Learning: A Survey (IJCAI, 2021)
> - Semenoglou et al. Data augmentation for univariate time series forecasting with neural networks (Pattern Recognition, 2023)
>
> > ### [Summary]: "They use augmentation strategies (Time Warping, Interpolation, and Noise Injection) to construct the different views".
>
> It seems there is misunderstanding on what multi-view is in our setup. We employ multi-scale patch division with various patch sizes as multi-views, rather than augmentation strategies. This is clearly depicted in **Figure 1** and described in **lines 162–166 of the paper**.
>
> This clarification also relates to your question [Q6]: “... different resolutions (scales)? …”
>
> We believe that different patch sizes correspond to different temporal resolutions in the time series. The dual attention mechanism in each individual-view model captures temporal dependencies at these specific resolutions. By aggregating information from various scales, the architecture effectively produces a comprehensive prediction.
>
> While ensembling complementary forecasts is a common approach in multi-scale modeling, our work addresses a different aspect of multi-scale (multi-view) modeling. When encountering unseen data, individual-view models may produce divergent predictions due to variability in single-scale representations. To address this, we enhance agreement among multiple single-view models by leveraging augmented unseen data. Our analysis in **Appendix A.2: Variance of Individual-View Model** demonstrates that applying consensus promotion improves the consistency between individual-view models compared to MRes (SL).

---

> ### Author Response · Authors · 2024-11-18
> **Response to Reviewer GeBU (Part 2)**
>
> Below, we address each of your comments and questions in detail.
>
> > ### [W2] In lines 031-033, the statement about the superiority of transformers in forecasting is misleading.
>
> Thank you for the reviewer’s valuable feedback. To provide a more comprehensive context for time series forecasting research, we have revised the **Introduction** and **Related Works** sections in the revised paper, with changes highlighted in blue.
>
>
> > ### [W3] The comparison against relevant baselines is lacking.
>
> - To address this feedback, we have included additional evaluation results in **Appendix A.5: Additional Baselines** to better contextualize current progress in long-term time series forecasting. Several advanced baselines have been added, with results sourced from their respective original papers. When compared to these state-of-the-art methods, CoMRes demonstrates favorable performance across the evaluated datasets.
> - However, we want to highlight that the primary objective of our experiments is to evaluate the most effective ways to utilize consensus promotion and augmented data, rather than to directly compare our method with other models. To clarify this, we have rephrased the **Baselines section in Main Text 4.1**. Consequently, although we have provided the requested comparisons upon the reviewer request, we believe that a direct comparison with other architectures is not strictly necessary.
> - As a potential extension of our proposed method, we recognize the possibility of applying consensus promotion on augmented data to other relevant architectures, such as TimeMixer. We acknowledge that applying our method to other architectures can further strengthen our paper and leave that as a future work in **Appendix A.9: Limitations and Future Work**.
>
> > ### [W4] The tables, particularly Table 1, are difficult to read.I'm not sure what red or blue means at least its not stated in the caption.
>
> Thank you for the reviewer’s valuable feedback. In response, we have revised **the caption of Table 1** and clarified certain **parts of Main Text 4.1** in the revised paper. These changes have been highlighted in blue for ease of reference.
>
> > ### [W5] The baseline method achieves the best result multiple times without the proposed augmentations, which weakens the argument for using the augmentations. In general I'm pretty confused about what the idea of multiple resolutions vs augmentations is in this work.
> > ### [Q5] Could you clarify whether MRes is your method (with ablated components)? I would not count that as a baseline.
>
> We believe there may be some misunderstanding regarding the experimental setup. As shown in Table 1, the first three columns (CoMRes) represent our proposed method, while the remaining columns—MRes w. augmentation, MRes w. consensus, MRes (SL)— correspond to baseline methods. These methods are not an existing approach but rather a version derived from CoMRes, created specifically for the ablation study by isolating certain components of the framework. The blue-highlighted delta values indicate that CoMRes consistently outperforms the baselines.
>
> - **MRes w. consensus as a Baseline**: Although it is a version we created, "MRes w. consensus" serves as one of the baseline methods and is not part of the proposed CoMRes framework. This baseline applies consensus promotion exclusively to labeled data. When compared to our model, CoMRes consistently demonstrates better performance across datasets. (CoMRes (ours) vs. MRes w. consensus shows superiority in 6 out of 8 datasets, with one dataset yielding comparable results.)
>
> As noted, consensus promotion is expected to reduce the variance of individual-view models. However, it is not always advantageous for the prediction task. When applied to labeled data, consensus promotion introduces a risk of overfitting, which we believe explains the inconsistent improvement patterns observed with "MRes w. consensus."
>
> In contrast, CoMRes applies consensus promotion on augmented data without assigning corresponding labels. This approach enhances the model’s generalization ability and is evidenced by the consistently lower MSE scores achieved by CoMRes across all evaluated datasets.
>
> Following the reviewer’s suggestions, we have:
> 1) Rephrased the "Baselines" section in Main Text 4.1 for clarity.
> 2) Elaborated on the performance comparisons between CoMRes and the baseline methods in Section 4.1 ("Results").
>
> All revisions have been included in the revised paper, with changes highlighted in blue.

---

> ### Author Response · Authors · 2024-11-18
> **Response to Reviewer GeBU (Part 3)**
>
> > ### [Q1] Could you provide specific details about what was missed in the multi-resolution reconciliation methods used by Wang et al.?
>
> TimeMixer (Wang et al., 2024), concurrent work with ours, is one of the state-of-the-art methods for multi-scale modeling in time series.While we do not believe there are not necessarily missing concepts in the multi-resolution reconciliation methods employed by TimeMixer, our work addresses a different aspect of multi-scale modeling.
>
> TimeMixer focuses on multi-scale feature extraction through its architecture(Past-Decomposable-Mixing and Future-Multipredictor-Mixing), whereas our approach emphasizes enhancing consistency and generalization by leveraging consensus promotion on augmented data. This distinction highlights the complementary nature of the two methods while demonstrating how our framework effectively tackles the challenge of reconciling diverse representations, particularly in the context of unseen data.
>
>
> > ### [Q2]  you mention that augmentations are less effective in forecasting compared to classification and anomaly detection. Could you elaborate on why this is the case? Specifically, how does your chosen augmentations differ?
>
> While "labels" for augmented data can theoretically be constructed by shifting the time window, we believe this is not straightforward. Augmented data is artificial and unseen, and using shifted values as labels may introduce noise. Similarly, Wen et al. (IJCAI, 2021) report that while augmentation boosts performance in classification and anomaly detection, it can yield negative results for certain data in forecasting.
>
> Moreover, as demonstrated in our experimental results(Table 1), MRes w. augmentation (which uses augmented data with corresponding labels) often leads to a decline in prediction accuracy. This finding supports our belief that using augmented data in this manner is less effective and may even hinder performance.
>
> In this study, we utilize the same data augmentation technique differently, using it exclusively for consensus promotion, which has shown improvements in most of our experiments.
>
> ### Ref.
> Wen et al. Time Series Data Augmentation for Deep Learning: A Survey (IJCAI, 2021)
>
>
> > ### [Q3] Why do you assume a Euclidean distance to measure consensus between the aggregated prediction and each view? Would other measures, such as soft-DTW, work better?
> > ### [Q4]  more "complex" augmentation strategies
> > ### [W6] While the limited-resource scenario is a useful addition, it does not effectively demonstrate how the method performs on datasets with many variables (e.g., Traffic or Electricity). ...
>
> We are grateful for your constructive suggestions. We prioritized conducting thorough evaluations on diverse datasets to ensure the generalizability and robustness of our method, which we consider a crucial aspect of time series forecasting research.
>
> We acknowledge the potential of exploring alternative measures, such as soft-DTW(differentiable loss functions for time series), to evaluate consensus between aggregated predictions and individual view, as well as adopting more complex augmentation strategies. We have noted these as limitations of the current study and outlined them as directions for future work in the revised paper under **Appendix A.9: Limitations and Future Work**.
> Additionally, we recognize the need to evaluate the method's performance on datasets with many variables (e.g., Traffic or Electricity). However, due to resource constraints, we were unable to conduct experiments on these datasets. This limitation has been explicitly addressed in **Section 4.3.1** of the main text in the revised paper.
>
> > ### [Q7] Could you clarify how exactly robustness was demonstrated?
>
> The robustness of CoMRes compared to the baselines is demonstrated through two key aspects:
>
> 1. **Better Performance on Test Sets**:
>    As shown in **Table 1**, CoMRes consistently outperforms the baselines across most datasets, highlighting its superior generalization ability.
>
> 2. **Reduced Error Propagation**:
>    By leveraging augmented unseen data and applying unsupervised consistency loss, CoMRes mitigates error propagation during training, leading to more stable predictions. As shown in **Figure 3**, while prediction error naturally increases along the horizon due to the intrinsic dynamics of time series, our proposed model consistently outperforms the baseline models, especially over longer horizons. Additionally, the results in **Section 4.2: Error Accumulation in Autoregressive Forecasting** further demonstrate CoMRes's superior predictive accuracy and reduced error accumulation.
>
> These factors collectively highlight the superior robustness of CoMRes in addressing the challenges of long-term time series forecasting.
>
> Thank you again for your feedback! We hope our explanations adequately address your questions and concerns. If not, please do not hesitate to let us know—we would be happy to discuss further.

---

> > ### Comment · Reviewer_GeBU · 2024-11-21
> > **Thank you + Remaining Questions**
> >
> > Thank you very much for your response and the clarifications you've provided.
> >
> > Regarding your response to W5 and Q5, I still find it confusing to label an ablation as a "baseline." As you correctly describe, the ablation is derived from your proposed method, which makes calling it a "baseline" misleading. In my opinion, this may give readers the wrong impression of how the comparisons were established.
> >
> > In general, while the results are quantitative, they empirically appear to be more on the qualitative side, as you only present them for a single model and ablate on a single augmentation. This does not empirically prove the feasibility of your method.
> >
> > i.e.line 303: *MRes w. augmentation: MRes Trained on labeled dataset that is augmented with Time Warping.*
> >
> > This makes it challenging to assess the genuine improvement brought by your framework.
> >
> > For instance, if you look at iTransformer [3], they did an excellent job in showing how their proposed method enhances other models (Table 2) and clearly laid out their ablations in Table 6.
> >
> >
> > Regarding Appendix A.2: Variance of Individual-View Model and Figure 3, could you clarify what the x-axis represents? Specifically, what is order?
> >
> > Regarding resource constraints, if increasing the variates  poses a significant bottleneck, it might be worthwhile to explicitly outline your resource footprint, including the required hardware. This would provide a clearer picture of the practical limitations involved.
> >
> >
> > Finally, I am still not entirely convinced by the downsides involving augmentations in a "supervised" scenario. I reviewed the two works you mentioned, and I would like to highlight some concerns. Wen et al. (2021) [1] evaluated augmentations such as cropping (which is quite distinct), flipping (also a significant transformation), and some frequency-based augmentation and Time warping. Even then, the performance only slightly decreased in 2 datasets, with one exhibiting a more significant drop. Moreover, these evaluations were conducted only on a Vanilla Transformer and DeepAR. While this provides some insights, I do not believe it offers sufficient empirical proof for your argument. They also mention:
> >
> > *"We observe that the data augmentation methods bring promising results for all models in average sense"* [1]
> >
> > *"However, the negative results can still be observed for specific data/model pairs"* [1]
> >
> > I acknowledge that the authors provide a compelling incentive, but I do not think severe augmentations like flipping or cropping are directly comparable to your noise injection, for example.
> >
> > Semenoglou et al. (2023) [2] also point out a similar finding in Section 5, where they note that "supervised" augmentations significantly improve results, while also admitting that some augmentations are more effective than others. Given that, I would like to suggest that it would be highly beneficial to show results on a per-augmentation basis. The conclusions in [1, 2] suggest that the issue may not be as severe as claimed, and that the observed performance drop might only hold for specific augmentations. For example, I would expect that noise injection, being relatively subtle, should lead to an improvement rather than a degradation in performance. It might also be helpful to qualitatively illustrate the differences between a noise-injected sample and a normal one.
> >
> >
> >
> > If I have misjudged any of these aspects, I am open to revising my evaluation and raising my score accordingly.
> >
> >
> >
> > [1] Wen et al. (2021). Time Series Data Augmentation for Deep Learning: A Survey. IJCAI
> >
> > [2] Semenoglou et al. (2023). Data augmentation for univariate time series forecasting with neural networks. Pattern Recognition
> >
> > [3] Liu et al. (2024). iTransformer: Inverted Transformers Are Effective for Time Series Forecasting. ICLR

---

> > > ### Author Response · Authors · 2024-11-23
> > > **Response II to Reviewer GeBU (Part 1)**
> > >
> > > Thank you for your attention and for providing valuable feedback again.
> > >
> > > > ### "...Given that, I would like to suggest that it would be highly beneficial to show results on a per-augmentation basis"
> > >
> > > Thank you for your insightful suggestion regarding the per-augmentation analysis.
> > > As shown in the table below, we conducted a per-augmentation analysis, comparing the performance of CoMRes with MRes w. augmentation across three augmentation strategies: Time Warping, Interpolation, and Noise Injection. While variability exists depending on the augmentation strategy, the overall results demonstrate that CoMRes consistently outperforms MRes w. augmentation across various augmentations and datasets.
> > >
> > > |         |      | Time Warp |       |         | Interpolation |       |         | Noise Injection |       |         | MRes(SL) |
> > > |:-------:|:----:|:---------:|:-----:|:-------:|:-------------:|:-----:|:-------:|:---------------:|:-----:|:-------:|:--------:|
> > > |         |      | CoMRes    | MRes  |Δ  | CoMRes        | MRes  | Δ  | CoMRes          | MRes |  Δ            |
> > > | **ETTh1**   | **96**   |   0.378   | 0.406 | 0.028   |     0.379     | 0.414 | 0.035   |      0.379      | 0.416 | 0.037   |   0.382  |
> > > |         | **192**  |   0.436   | 0.455 | 0.019   |     0.436     | 0.459 | 0.023   |      0.436      | 0.44  | 0.004   |   0.439  |
> > > |         | **336**  |   0.452   | 0.482 | 0.030   |     0.452     | 0.448 | -0.004  |      0.452      | 0.446 | -0.006  |   0.451  |
> > > |         | **720**  |   0.474   | 0.475 | 0.001   |     0.478     | 0.473 | -0.005  |      0.478      | 0.447 | -0.031  |   0.481  |
> > > |         | **Avg.** |   0.435   | 0.455 | 0.020   |     0.436     | 0.449 | 0.012   |      0.436      | 0.437 | 0.001   |   0.438  |
> > > | **ETTh2**   | **96**   |   0.275   | 0.278 | 0.003   |     0.275     | 0.28  | 0.005   |      0.275      | 0.28  | 0.005   |   0.277  |
> > > |         | **192** |   0.347   | 0.355 | 0.008   |     0.346     | 0.355 | 0.009   |      0.346      | 0.352 | 0.006   |   0.348  |
> > > |         | **336**  |   0.326   | 0.346 | 0.020   |     0.328     | 0.335 | 0.007   |      0.326      | 0.336 | 0.010   |   0.332  |
> > > |         | **720**  |   0.393   | 0.404 | 0.011   |     0.398     | 0.402 | 0.004   |      0.397      | 0.402 | 0.005   |   0.406  |
> > > |         | **Avg.** |   0.335   | 0.346 | 0.011   |     0.337     | 0.343 | 0.006   |      0.336      | 0.343 | 0.007   |   0.341  |
> > > | **ETTm1**   | **96**  |   0.310   | 0.326 | 0.016   |     0.311     | 0.418 | 0.107   |      0.311      | 0.418 | 0.107   |   0.312  |
> > > |         | **192**  |   0.362   | 0.367 | 0.005   |     0.364     | 0.43  | 0.066   |      0.364      | 0.436 | 0.072   |   0.361  |
> > > |         | **336** |   0.386   | 0.404 | 0.018   |     0.386     | 0.456 | 0.070   |      0.387      | 0.518 | 0.131   |   0.381  |
> > > |         | **720**  |   0.456   | 0.467 | 0.011   |     0.456     | 0.504 | 0.048   |      0.456      | 0.532 | 0.076   |   0.454  |
> > > |         | **Avg.** |   0.379   | 0.391 | 0.013   |     0.379     | 0.452 | 0.073   |      0.380      | 0.476 | 0.097   |   0.377  |
> > > | **ETTm2**   | **96**   |   0.163   | 0.168 | 0.005   |     0.163     | 0.175 | 0.012   |      0.163      | 0.176 | 0.013   |   0.167  |
> > > |         | **192** |   0.229   | 0.230 | 0.001   |     0.229     | 0.24  | 0.011   |      0.229      | 0.241 | 0.012   |   0.231  |
> > > |         | **336** |   0.291   | 0.295 | 0.004   |     0.292     | 0.308 | 0.016   |      0.292      | 0.308 | 0.016   |   0.292  |
> > > |         | **720** |   0.369   | 0.385 | 0.016   |     0.369     | 0.405 | 0.036   |      0.369      | 0.401 | 0.032   |   0.369  |
> > > |         | **Avg.** |   0.263   | 0.270 | 0.007   |     0.263     | 0.282 | 0.019   |      0.263      | 0.282 | 0.018   |   0.265  |
> > > | **Weather** | **96**   |   0.151   | 0.155 | 0.004   |     0.151     | 0.159 | 0.008   |      0.151      | 0.163 | 0.012   |   0.152  |
> > > |         | **192**  |   0.199   | 0.203 | 0.004   |     0.200     | 0.205 | 0.005   |      0.200      | 0.208 | 0.008   |   0.199  |
> > > |         | **336**  |   0.244   | 0.248 | 0.004   |     0.246     | 0.25  | 0.004   |      0.246      | 0.252 | 0.006   |   0.245  |
> > > |         | **720**  |   0.335   | 0.334 | -0.001  |     0.335     | 0.334 | -0.001  |      0.334      | 0.338 | 0.004   |   0.334  |
> > > |         | **Avg.** |   0.232   | 0.235 | 0.003   |     0.233     | 0.237 | 0.004   |      0.233      | 0.240 | 0.008   |   0.233  |
> > > #### **Table**. Results of per-augmentation analysis (MSE).  Δ represents the difference between MRes w. augmentation and CoMRes for each augmentation strategy.

---

> ### Author Response · Authors · 2024-11-23
> **Response II to Reviewer GeBU (Part 2)**
>
> (continue)
> The results may depend on the specific augmentations chosen and their associated hyperparameters (e.g., the variance of injected noise). However, it is important to highlight that the same augmentation can yield different outcomes depending on whether it is applied in a supervised or semi-supervised context. In supervised scenarios (e.g., MRes w. augmentation), augmentations may result in degraded performance due to overfitting or mismatched distributions. Conversely, in semi-supervised contexts (e.g., CoMRes), augmentations improve generalization by leveraging the unlabeled data more effectively.
> This observation supports our key finding that leveraging augmented data in a semi-supervised setting enhances prediction accuracy and generalization, even across varying augmentation strategies. (**Appendix A.8 Discussion - Comparison on a Per-Augmentation Basis** in the revised paper)
>
> > ### "Regarding Appendix A.2: Variance of Individual-View Model and Figure 3, could you clarify what the x-axis represents? Specifically, what is order?"
>
> Thank you for pointing this out. The x-axis represents the prediction horizon order. Specifically:
> - The x-axis corresponds to the prediction steps, with each step representing the forecasted value at that particular horizon.
> - For example, when h=96, the x-axis spans from 1 to 96, where **1** indicates the prediction for **the very next future value immediately following the lookback window**, and **96** represents **the farthest predicted value** in the horizon.
>
>
> This alignment along the prediction horizon allows us to observe how the error (MSE) accumulates as the forecast extends further into the future in Figure3.
>
> > ### " I still find it confusing to label an ablation as a "baseline." As you correctly describe, the ablation is derived from your proposed method, which makes calling it a "baseline" misleading. "
>
>
> We understand that labeling something we developed as a "baseline" may cause confusion. To clarify, our true baseline is Pathformer, a state-of-the-art long-term forecasting model that outperforms most other models, such as TimeMixer, NLinear, and others (as shown in Appendix Table 8). CoMRes consistently demonstrates superior performance compared to Pathformer, as highlighted in the table below:
>
> |             | CoMRes | Pathformer | PatchTST | TimeMixer | Nlinear | Scaleformer | TiDE  |
> |-------------|--------|------------|----------|-----------|---------|-------------|-------|
> | ETTh1       | 0.435  | 0.439      | 0.455    | 0.447     | 0.448   | 0.447       | 0.518 |
> | ETTh2       | 0.335  | 0.344      | 0.367    | 0.365     | 0.382   | 0.449       | 0.387 |
> | ETTm1       | 0.379  | 0.382      | 0.384    | 0.381     | 0.402   | 0.466       | 0.413 |
> | ETTm2       | 0.263  | 0.273      | 0.283    | 0.275     | 0.281   | 0.302       | 0.293 |
> | Electricity | 0.177  | 0.182      | 0.205    | 0.182     | 0.206   | 0.203       | 0.209 |
> | Traffic     | 0.490  | 0.501      | 0.507    | 0.485     | 0.624   | 1.097       | 0.608 |
> | Weather     | 0.232  | 0.239      | 0.257    | 0.24      | 0.251   | 0.436       | 0.271 |
>
>
> #### **Table**. Multivariate time series forecasting results (MSE). All results are averaged across 4 different prediction lengths: {96, 192, 336, 720}. Results for TimeMixer are sourced from Wang et al. (2024), PatchTST from Nie et al. (2023), and the remaining results from Chen et al. (2024). Full results are available in Appendix A.5 (Table 8).
>
> While it is true that the ablations are derived from CoMRes, we included them as baselines for practical purposes—to systematically compare and isolate the contributions of key components in our framework. This allowed us to thoroughly analyze the core strategies of CoMRes, specifically consensus promotion and semi-supervised learning using augmented unlabeled data.
> To address this concern, we will:
> - Explicitly clarify in the revised manuscript that these **ablations are derived from CoMRes for ablation study purposes, not independent baselines,** to avoid any potential misunderstanding.
> - Rephrase the **"Baseline"** section and add an **"Ablations"** section in Main Text 4.1 for greater clarity.
>
> We hope these revisions resolve the issue and provide clearer insights into our experimental design and results.
>
> > ### low resource experiment on datasets with many variables and resource constraints
>
> To address this, we are currently conducting limited-resource experiments on the Traffic dataset, **spanning 4 full days of experiments using 6 H100 GPUs with 80GB of memory** each. We expect to have the results available during the rebuttal period.
> We acknowledge the importance of transparency regarding the resource constraints. In the revised paper, we will explicitly outline the resource footprint of our experiments.
>
>
> Thank you again for your feedback! We hope our explanations adequately address your questions and concerns.

---

> > ### Comment · Reviewer_GeBU · 2024-11-25
> > **Last Concern**
> >
> > I thank the authors again for their clarifications and the newly introduced experiments.
> > I am still not entirely convinced that augmentations truly introduce a drawback (as the authors clearly stated, it depends on the exact configuration). However, I acknowledge that in your setting this seems mostly to be the case and I believe that the authors, given the limitations, have produced interesting work. I am inclined to raise my score but remain a bit confused about the last part of your response.
> >
> > > spanning 4 full days of experiments using 6 H100 GPUs with 80GB of memory
> >
> > In your appendix, you state that all experiments were carried out using:
> > "All experiments are implemented in PyTorch and executed on an NVIDIA A6000 48GB GPU."
> >
> > Is evaluating on traffic such a limitation? Do datasets with increasing variety or size hinder the applicability of your method? Could you briefly state the training times (and perhaps the inference times) for the different datasets?

---

> ### Author Response · Authors · 2024-11-25
> **Response III to Reviewer GeBU**
>
> Thank you for your thoughtful feedback and for considering raising your score. We deeply appreciate your acknowledgment that our study has produced interesting work.
>
> > On concern about datasets with increasing variety or size
>
> We apologize for any misunderstanding caused. CoMRes, MRes, and Pathformer share a similar architecture; however, the training time of CoMRes is approximately 1.5 times longer, primarily due to the overhead introduced by the data loader when generating augmented data on-the-fly.
>
> Most of the experiments presented in the paper, including ablation studies, were conducted on an NVIDIA A6000 GPU (48GB). For the additional experiments, conducted specifically for the rebuttal discussion, we used H100 GPUs with 80GB memory (6 devices for running 6 experiments simultaneously) due to the time constraints of the rebuttal period. To clarify, the decision to use H100 GPUs was purely motivated by the tight rebuttal timeline -- and there is no other reason than that.
>
> **We sincerely apologize for any confusion caused—our intention was simply to emphasize the effort we put into providing the requested results and meeting the rebuttal deadline.**
>
> To address your request, we provide approximate training times of CoMRes under our standard experimental setup (an NVIDIA A6000):
> - Small datasets (e.g., ETTh1, ETTh2, ETTm1, ETTm2, Weather): ~1 hour per dataset.
> - Large datasets (e.g., Electricity, Traffic): 3~6 hours per dataset, primarily due to the computational cost of generating augmented data dynamically.
>
> Inference times are efficient, averaging 10–20 seconds for all test samples (and ~30 seconds for Traffic), depending on the sequence length and model configuration.
>
> We hope this clarification provides a clearer understanding of the computational resources required for our method. Please let us know if you have further questions or concerns.

---

> > ### Comment · Reviewer_GeBU · 2024-11-25
> >
> > Thank you for these clarifications and that you cleared the caused confusion. I would recommend to incorporate such information (i.e. runtime) in the final version of the paper. The increased training time is a limitation which limits the application of CoMRes.
> >
> > Thank you for the good rebuttal.
> >
> > I would like to increase my score to a borderline accept.

---

> > > ### Author Response · Authors · 2024-11-26
> > > **Thank you for your kind reply**
> > >
> > > We greatly appreciate your feedback. Your insightful and constructive comments have been invaluable in helping us enhance the quality of our work.
> > >
> > > Also, we will include approximate training and inference times in the updated version of our paper to provide a clearer understanding of the computational resources required.

---

### Official Review · Reviewer_2k6g · 2024-11-03

**Soundness:** 3
**Presentation:** 3
**Contribution:** 3
**Rating:** 8
**Confidence:** 3

**Summary:**

This paper addresses challenges in long-term time series forecasting, particularly in managing unseen patterns and data scarcity. The authors propose CoMRes, a semi-supervised, multi-view approach to enhance forecasting by promoting consensus among multiple models on augmented, unseen data. This method aims to improve prediction accuracy and reduce error accumulation over extended horizons. Additionally, the study examines both autoregressive and non-autoregressive decoding schemes, emphasizing the robustness of non-autoregressive decoding in minimizing long-term error propagation. Experimental results highlight CoMRes's superior performance and stability compared to traditional supervised models.

**Strengths:**

The strengths are as follows:

- The paper is clear, well-written and presented.
- The authors cover most of the commonly used datasets in the field, and some relevant baselines: the empirical constributions appear sound as a result.
- Results indicate the method does indeed provide some life over the baselines.

**Weaknesses:**

The weaknesses are as follows:

- Absence of submitted code making reproducibility more difficult.
- The paper could benefit from more detailed comparisons with other time-series forecarsting methods.
- It would be interesting to include error bars on the results table, especially given the proximity of different methods.

**Questions:**

- Could the authors add error bars?
- What are the author's plans wrt. the release of code?

Edit: the authors have addressed the questions and points I raised to a level that I find satisfactory. I am raising my score as a result.

---

> ### Author Response · Authors · 2024-11-18
> **Response to Reviewer 2k6g**
>
> Thank you for taking the time and providing thoughtful and valuable feedback. Below, we address each of your comments and questions in detail.
>
> > ### [Q1] “Could the authors add error bars?”
>
> Thanks for the reviewer's valuable suggestion. We have provided the standard deviation results in **Appendix A.4: Error bars**.
>
>
> > ### [Q2] “What are the author's plans wrt. the release of code?”
>
> We are still in the process of refactoring the code; however, we are providing the current version to ensure reproducibility. The code is available in an anonymous GitHub repository, accessible via the following link: [https://anonymous.4open.science/r/anonymous-9FB7/](https://anonymous.4open.science/r/anonymous-9FB7/).
>
> > ### [W2] “The paper could benefit from more detailed comparisons with other time-series forecasting methods.”
>
> - To address this feedback, we have included additional evaluation results in **Appendix A.5: Additional Baselines** to better contextualize current progress in long-term time series forecasting. Several advanced baselines have been added, with results sourced from their respective original papers. When compared to these state-of-the-art methods, CoMRes demonstrates favorable performance across the evaluated datasets.
>
> - The primary objective of our experiments is to evaluate the most effective ways to utilize consensus promotion and augmented data, rather than to directly compare our method with other models. To clarify this, we have rephrased **the "Baselines" section in Main Text 4.1**. Consequently, we believe that a direct comparison with other architectures is not strictly necessary.
>
> - As a potential extension of our proposed method, we acknowledge the possibility of applying consensus promotion on augmented data to other relevant architectures, such as TimeMixer. We have noted this as a limitation of the current study and outlined it as a direction for future work in the revised paper under **Appendix A.9: Limitations and Future Work**.
>
>
> Thank you again for your feedback!  We hope our explanations adequately address your questions and concerns. If not, please do not hesitate to let us know—we would be happy to discuss further.

---

> ### Author Response · Authors · 2024-11-23
> **Request of Reviewer's attention and feedback**
>
> Dear Reviewer,
>
> Thanks for your valuable and constructive review, which has inspired us to improve our paper further substantially. This is a kind reminder that it has been 3 days since we posted our rebuttal. Please let us know if our response has addressed your concerns.
>
> Following your suggestions, we have provided the following revisions to our paper:
> - A.4 Error bars
> - A.5 Additional baselines
> - Code for Reproducibility : https://anonymous.4open.science/r/anonymous-9FB7/
>
> Sincere thanks for your dedication! We are looking forward to your reply.

---

> > ### Comment · Reviewer_2k6g · 2024-11-28
> > **Response to rebuttal**
> >
> > I thank the authors for addressing my points. I have raised my score as a result as all my concerns were addressed.

---

> > > ### Author Response · Authors · 2024-11-28
> > > **Thank you for your kind reply**
> > >
> > > We sincerely thank the reviewer for acknowledging our efforts in addressing the raised points and for the kind gesture of increasing the score. We are pleased that our revisions have effectively resolved your concerns, and we greatly appreciate your thoughtful feedback and support in improving the quality of our manuscript.
> > >
> > > Please feel free to reach out if you have any further questions or concerns.
> > >
> > > Thank you again for your valuable time and consideration.

---

### Author Response · Authors · 2024-11-18
**Summary of Rebuttal Revisions**

We sincerely thank all the reviewers for their insightful reviews and valuable comments, which are instructive for us to improve our paper further.

This paper proposes **CoMRes**, a semi-supervised framework that leverages consensus promotion on augmented data without requiring explicit future values as labels. Our approach addresses a different aspect of multi-scale (multi-view) modeling and long-term time series forecasting (LSTF) compared to existing studies, **focusing on improving consistency and generalization when encountering unseen data**. Individual-view models often produce divergent predictions due to variability in single-scale representations. To address this, CoMRes enhances agreement among multiple single-view models by leveraging augmented unseen data.

Addressing the effective utilization of unlabeled data in time series forecasting is a relatively unexplored yet crucial area, and we believe our approach paves the way for further research.

We have made every effort to address all concerns by providing detailed clarifications and requested results. Below is a summary of the major revisions:

**1. Main text** - All revisions have been incorporated in the revised paper, with changes highlighted in blue.
- **introduction and related works** : Revised to better articulate our contributions, with changes highlighted in blue.
- **4.1 Time Series Forecasting**: To highlight the primary objective of our experiments
  - Rephrased the "Baseline" section for clarity.
  - Add the "Ablations" section for clarity.
  - Elaborated on the performance comparisons between CoMRes and the baseline methods in "Results" section and Table 1.
- **4.3.1 Limited-Resource Scenarios** : Add Traffic result to demonstrate how the method performs on datasets with many variables

**2. Appendix** - Following the reviewers' suggestions, we have included the following:
- A.2 Variance of Individual-view Model (**Reviewer iocg**)
- A.4 Error bars (**Reviewer 2k6g, iocg**)
- A.5 Additional baselines (**Reviewer 2k6g, GeBU**)
- A.7 Discussion - More Related Works (**Reviewer GunG**)
- A.8 Discussion - Result on ILI dataset (**Reviewer GunG**)
- A.9 Discussion - Comparison on a Per-Augmentation Basis (**Reviewer GeBU**)
- A.10 Discussion - Computation Cost (**Reviewer GeBU, GunG**)
- A.12 Limitation and Future Work  (**Reviewer GeBU**)

**3. Code for Reproducibility** (Reviewer 2k6g):
The code has been made available at the following anonymous repository: [https://anonymous.4open.science/r/anonymous-9FB7/](https://anonymous.4open.science/r/anonymous-9FB7/)

We sincerely hope that the above revisions adequately address the reviewers' concerns and provide clarity, thereby strengthening the soundness and highlighting the contributions of our paper.

---

### Author Response · Authors · 2024-12-03
**General Response and Revision Summary**

We sincerely thank all the reviewers for their thoughtful feedback and constructive comments. Your insights have been invaluable in helping us refine and improve the quality of our submission.

We greatly appreciate the reviewers’ positive recognition of our work. CoMRes has been acknowledged for introducing an interesting and **novel perspective of promoting consistency** among multi-resolution views, **representing a promising direction in time-series forecasting (GeBU, iocg)**. The paper addresses **an important direction of multi-resolution time-series forecasting(GunG)**, a critical area for advancing temporal modeling techniques. The sound empirical contributions, **in-depth ablation studies, extensive datasets, and inclusion of limited-resource scenarios(2k6g, GeBU, GunG)** further provide valuable insights into the method’s effectiveness, enhancing credibility and practical value for real-world applications. Additionally, we are pleased that **the clarity and presentation of our work were well-received. (2k6g, GeBU)**.

In response to the insightful and constructive concerns raised, we have made every effort to address them comprehensively and incorporated all suggested improvements. Key revisions include:
- **Enhancing Evaluation and Analysis:**
  - Added comparisons with additional baselines to contextualize current progress (Appendix A.5) (2k6g, GeBU).
  - Included error bars and statistical significance tests to rigorously evaluate performance variability (Appendix A.4) (2k6g, iocg).
  - Conducted variance analysis of multi-view models to examine the effectiveness of the unsupervised loss function (Appendix A.2) (iocg).
  - Detailed per-augmentation analysis to address varying augmentation strategies (Appendix A.8) (GeBU).
  - Expanded discussion on computation costs, including runtime comparisons, to clarify computational efficiency (Appendix A.10) (GeBU, GunG).
  - Added results on the Traffic dataset to demonstrate performance on datasets with many variables (Section 4.3.1, Limited-Resource Scenarios) (GeBU).
- **Clarifications:**
  - Revised the introduction and related works section to articulate contributions more effectively (GeBU, iocg, GunG).
  - Expanded the discussion of related works in Appendix A.7 (GunG).
  - Rephrased the “Baseline” section in Section 4.1, added an “Ablations” section, and elaborated on performance comparisons between CoMRes and ablations in the "Results" section and Table 1 (GeBU).
  - Included discussions on the ILI dataset (Appendix A.8) and limitations/future work (Appendix A.12) to address additional feedback (GeBU, GunG).

All revisions have been highlighted in blue in the revised manuscript. **We are sincerely grateful for the reviewers’ constructive engagement, which has greatly enhanced the quality of this work.** We appreciate the reviewers' recognition of our efforts and increased scores reflecting the improvements.

---

### Meta-Review · Area_Chair_9dqQ · 2024-12-15

**Metareview:**

The paper introduces CoMRes, a semi-supervised time series forecasting framework that promotes consensus across multi-resolution augmented data views without requiring explicit labels. The authors claim this method improves forecasting accuracy and robustness, particularly in long-term predictions, while mitigating error accumulation. Strengths include a novel application of consensus promotion to time series forecasting, robust empirical evaluation across diverse datasets, and effective handling of limited-resource scenarios. However, weaknesses include limited methodological novelty as the framework adapts existing approaches like Pathformer, and some reviewers raised concerns about computational overhead and insufficient clarity on the generalization of augmentations. Missing elements include direct comparisons with advanced baselines and deeper analysis of computational trade-offs. While the method demonstrates practical utility and robust results, the concerns about novelty and computational efficiency contributed to mixed reviews, but overall improvements during the rebuttal and strong empirical performance led to a borderline acceptance recommendation.

**Additional Comments On Reviewer Discussion:**

During the rebuttal period, reviewers raised concerns about CoMRes’s novelty, computational efficiency, and the effectiveness of its consensus promotion strategy. Specific issues included comparisons to existing methods like Pathformer, the need for detailed variance analysis, and computational overhead from augmented data. The authors responded by providing additional baselines, variance analyses, p-value tests, and runtime metrics, while clarifying the distinct challenges addressed by their approach. They demonstrated improved results across datasets and clarified design choices in the manuscript. While reviewers appreciated the authors’ efforts and resolved some concerns, skepticism about novelty and computational trade-offs remained. Ultimately, the strong empirical results and thorough revisions influenced a borderline acceptance decision, emphasizing the practical contributions despite lingering questions about originality.

---

### Decision · Program_Chairs · 2025-01-22

Accept (Poster)